# FlexSBDD: Structure-Based Drug Design with Flexible Protein Modeling

**Zaixi Zhang**[1,2,3], **Mengdi Wang**[3], **Qi Liu**[1,2]*

1: School of Computer Science and Technology, University of Science and Technology of China
2: State Key Laboratory of Cognitive Intelligence, Hefei, Anhui, China
3: Princeton University
zaixi@mail.ustc.edu.cn, mengdiw@princeton.edu, qiliuql@ustc.edu.cn

## Abstract

Structure-based drug design (SBDD), which aims to generate 3D ligand molecules binding to target proteins, is a fundamental task in drug discovery. Existing SBDD methods typically treat protein as rigid and neglect protein structural change when binding with ligand molecules, leading to a big gap with real-world scenarios and inferior generation qualities (e.g., many steric clashes). To bridge the gap, we propose FlexSBDD, a deep generative model capable of accurately modeling the flexible protein-ligand complex structure for ligand molecule generation. FlexSBDD adopts an efficient flow matching framework and leverages E(3)-equivariant network with scalar-vector dual representation to model dynamic structural changes. Moreover, novel data augmentation schemes based on structure relaxation/sidechain repacking are adopted to boost performance. Extensive experiments demonstrate that FlexS-BDD achieves state-of-the-art performance in generating high-affinity molecules and effectively modeling the protein's conformation change to increase favorable protein-ligand interactions (e.g., Hydrogen bonds) and decrease steric clashes.

## 1 Introduction

Deep generative models are profoundly impacting drug discovery, particularly within the challenging subfield of structure-based drug design (SBDD) [1, 54, 36]. SBDD focuses on generating drug-like ligand molecules conditioned on target-binding proteins, necessitating precise modeling of complex geometric structures and detailed protein-ligand interactions. Some early attempts adopt autoregressive models to generate 3D ligand molecules atom-by-atom [53, 57] or fragment-by-fragment [85]. To overcome the limitations of autoregressive methods (e.g., error accumulation), recent works [28, 29] leverage non-autoregressive diffusion-based models [31] to predict the distribution of ligand atom types and positions via denoising and have achieved the state-of-the-art performance.

Despite the remarkable success, most existing SBDD models treat target proteins as rigid and neglect the conformation change. However, according to the "induced fit" theory in biochemistry [43], proteins are flexible structures that undergo structural changes upon ligand binding, leading to enhanced interactions and binding affinity. In structural biology, the protein structure that has a bound small molecule is referred to as ligand-bound or **holo** conformation, and the protein structure without a bound small molecule is called ligand-free or **apo** conformation [16]. For example, Figure. 1 (a)&(b) shows the aligned apo and holo-structures of two proteins, and the extent of structural change is influenced by the specific properties and structures of the proteins and ligands. The neglect of protein flexibility in SBDD leads to several significant drawbacks: (1) The generated protein-ligand complexes are prone to have sub-optimal protein structures and steric clashes as the protein cannot adaptively adjust structures according to different generated molecules [30]. (2) The chemical search

---

*Qi Liu is the corresponding authors.

space is overly restricted by existing holo data, limiting the exploration of diverse high-quality drugs. As a result, SBDD models tend to produce molecules that are similar to those fitting the predefined pocket space [53, 62]. (3) A large gap between real-world physical binding process and computational simulations, which may lead to high false positive rates in real-world drug discovery applications i.e., most computationally designed drugs do not have real therapeutic effects [2].

However, several challenges exist for flexible protein modeling in existing SBDD models. Firstly, proteins are macromolecules with thousands of atoms, which brings formidable high degrees of freedom for flexible structural modeling [35]. Secondly, there lack apo-holo structure pair datasets for learning structural changes [16]. Finally, the widely used diffusion-based models are time-consuming for exploring the huge space of flexible protein-ligand complexes [28].

To address the aforementioned challenges, we propose a new method FlexSBDD capable of modeling the flexibility of target protein while generating *de novo* 3D ligand molecules. To reduce the computational complexity, we focus on the key degrees of freedom in protein structure (i.e., $C_\alpha$ coordinate, the orientation of the backbone frame, and the sidechain dihedral angels to determine the full atom structure) inspired by previous works [67, 77, 9]. As for the dataset, we employ the Apobind dataset [3] with additional Apo data generated by OpenMM relaxation [21] and Rosetta repacking [18] as data augmentation. For efficient and stable generation, we adopt a flow-matching framework [49, 5] that defines multimodal conditional flows for different components in protein-ligand complex and use E(3)-equivariant network with scalar-vector dual representation for learning the chemical and geometric information. The input to FlexSBDD is the initialized protein structure (e.g., apo structure). FlexSBDD learns to iteratively update protein-ligand structures and ligand atom types from $time = 0$ to $1$ and finally outputs both the generated 3D ligand molecule and the updated protein structure (holo). Extensive evaluations on benchmarks and case studies show the advantage of FlexSBDD in generating structurally valid protein-ligand complexes with high affinity, more favorable non-covalent interactions, and fewer steric clashes. The code of the paper is provided at `https://github.com/zaixizhang/FlexSBDD`. We highlight our main contributions as follows:

- We propose a flow-matching-based generative model FlexSBDD, capable of modeling protein flexibility while generating *de novo* 3D ligand molecules.

- FlexSBDD not only achieves state-of-the-art performance on benchmark datasets (e.g., -9.12 Avg. Vina Dock score), but also learns to adjust the protein structure to increase favorable interactions (e.g., 1.96 Avg. Hydrogen bond acceptors) and decrease steric clashes.

- With a concrete case study on KRAS[G12C], a promising target of solid tumor, we demonstrate FlexSBDD's potential to discover cryptic pockets for drug discovery.

## 2   Related Works

### 2.1   Structure-Based Drug Design

Structure-based drug design (SBDD) [87] aims to directly generate 3D ligand molecules inside target protein pockets. LiGAN [62] first uses 3D CNN to encode the protein-ligand structures and generate ligands by atom fitting and bond inference from the predicted atom densities. Several follow-up works have adopted autoregressive models for atom-wise [53, 50, 57, 79, 78] or fragment-wise [27, 60, 85, 78] generation of 3D molecules. For example, Pocket2Mol [57] adopts the geometric vector perceptrons [40] as the context encoder and autoregressively predicts the atom types, atom coordinates, and bond types until the generated molecule is completed. FLAG [85] and DrugGPS [82] predict the next molecular fragment and add it to the partially generated molecule in each round. Recently, powerful diffusion models have started to make a significant impact in SBDD, demonstrating promising results with non-autoregressive sampling [65, 28, 29, 46]. They usually represent the protein-ligand complex as 3D atom point sets, and define diffusion and denoising processes for both continuous atom coordinates and discrete atom types. For instance, TargetDiff [28] proposes a target-aware molecular diffusion process with a SE(3)-equivariant GNN denoiser. While there has been significant progress, existing approaches often neglect the critical aspect of protein flexibility. To address this gap, we explicitly model the conformational change of protein in FlexSBDD.

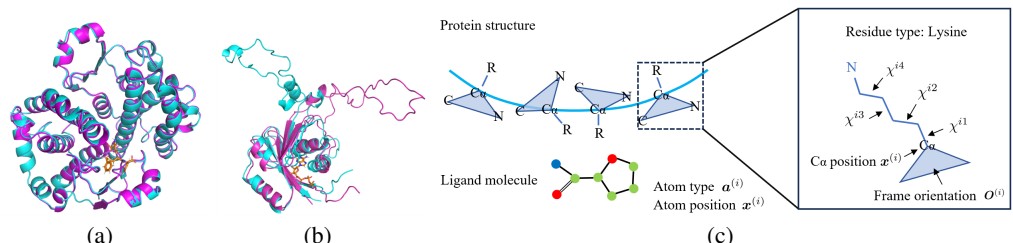

Figure 1: Aligned apo (ligand-free) and holo (ligand-bound) examples of (a) Human Glutathione S-Transferase protein (PDB ID: 10GS) and (b) Human Menkes protein (PDB ID: 2KMX) [3]. Apo structures are colored in "magenta" and holo-structures in "cyan" with the bounded ligand in "orange". (c) Illustration of the protein structure and ligand molecule parameterization.

## 2.2 Flexible Protein Modeling

Proteins are intrinsically dynamic entities and flexibility is a critical factor affecting protein's function and behavior in biological systems [72, 37]. The dynamic properties are traditionally modeled with physics-based methods such as Molecular Dynamics simulation (MD) [33]. To overcome the computationally demanding drawback of MD, some deep learning-based methods have been recently proposed, e.g., DynamicBind [51], FlexPose [20], SBAlign [68], NeuralPlexer [61], and DiffDock-Pocket [59] consider protein flexibility in protein-ligand docking. However, these methods can hardly extended to the challenging *de novo* ligand generation, leaving it an unsolved problem.

## 3 Preliminaries

**Notations and Problem Formulation:** We model protein-ligand complex as $\mathcal{C} = \{\mathcal{P}, \mathcal{G}\}$ [84, 86, 76, 83]. The objective of FlexSBDD is to learn a conditional generative model $p(\{\mathcal{P}', \mathcal{G}\}|\mathcal{P})$ that generates ligand molecule conditioned on the target protein and meanwhile updates the structure of the target protein ($\mathcal{P}'$). Proteins are composed of a sequence of residues (amino acids), each containing 4 backbone atoms (i.e., $C_\alpha, N, C, O$) and a sidechain that identifies the residue type. In this paper, the residue types are assumed known and the full atom structure of an amino acid can be represented by its $C_\alpha$ coordinates $\boldsymbol{x}_i \in \mathbb{R}^3$, the frame orientation $\boldsymbol{O}^{(i)} \in SO(3)$, and maximally 4 sidechain dihedral angles $\boldsymbol{\chi}^{(i)} = \{\chi^{i1}, \chi^{i2}, \chi^{i3}, \chi^{i4}\} \in [0, 2\pi)^4$, where $i \in \{1, \cdots N_p\}$ and $N_p$ is the number of residues in a protein. The backbone structure of the residue can be determined according to their ideal local coordinates relative to the $C_\alpha$ position $\boldsymbol{x}^{(i)}$ and the orientation $\boldsymbol{O}^{(i)}$ [23]. The sidechain conformations can be derived with the dihedral angles $\boldsymbol{\chi}^{(i)}$ as the bond length/angles are largely fixed [81]. With the above notations, a protein structure with $N_p$ residues can be compactly represented as $\mathcal{P} = \{\boldsymbol{x}^{(i)}, \boldsymbol{O}^{(i)}, \boldsymbol{\chi}^{(i)}\}_{i=1}^{N_p}$ (see the illustration in Figure. 1(c)). Following previous works [74], we treat each amino acid as a node and integrate backbone orientation and sidechain dihedral angles as node features. Such method enjoys the advantage of fewer nodes and less computational cost. The generated ligand molecule can be represented as a set of atoms: $\mathcal{G} = \{\boldsymbol{a}^{(i)}, \boldsymbol{x}^{(i)}\}_{i=1}^{N_l}$, where $\boldsymbol{a}^{(i)} \in \mathbb{R}^{n_a}$ indicate the atom type ($n_a$ is the total number of atom types we consider) and $\boldsymbol{x}^{(i)} \in \mathbb{R}^3$ denotes the atom coordinate. In the constructed protein-ligand complex 3D graph, we use $\boldsymbol{x}^{(i)}$ to denote both $C_\alpha$ coordinates and the ligand atom coordinates for conciseness.

**Riemanian Flow Matching:** Flow Matching (FM) [49, 5, 4], a simulation-free method for learning continuous normalizing flows (CNFs) [15], has shown better performance and efficiency than diffusion-based models on a series of biomolecular tasks [9, 77, 69]. To model the complicated protein-ligand complex structures, we need to apply the general flow matching on the Riemannian manifolds [14]. Let $\mathcal{M}$ be the manifold space with metric $g$. $q$ is the probability distribution of data $x \in \mathcal{M}$, and $p$ be the prior distribution. The time-dependant probability path on $\mathcal{M}$ is defined as $p_{t \in [0,1]} : \mathcal{M} \to \mathbb{R}_{>0}$ satisfying $p_0 = p$ and $p_1 = q$. $u_t(x) \in \mathcal{T}_x\mathcal{M}$ is the corresponding gradient vector of the path on $x$ at time $t$. Flow Matching aims to learn a neural network $v_\theta(x, t)$ to approximate the target vector field $u_t$: $\mathcal{L}_{FM}(\theta) = \mathbb{E}_{t, p_t(x)} \|v_\theta(x, t) - u_t(x)\|_g^2$. However, $u_t$ is intractable in practice and an alternative is to use a conditional density path $p_t(x|x_1)$ with a conditional gradient

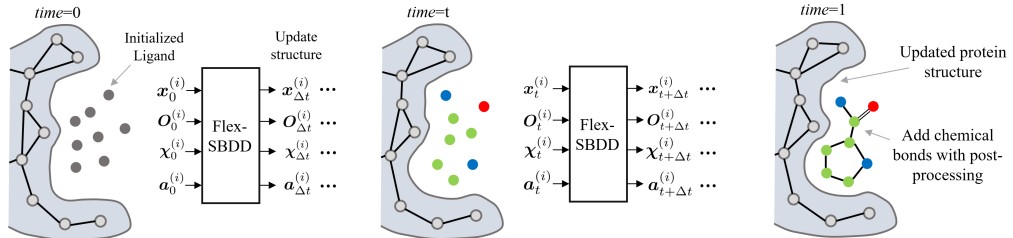

Figure 2: **Overview of FlexSBDD**. The flow matching-based generative process starts from an apo protein structure and the initialized ligand molecule. At each time step, FlexSBDD updates $\mathcal{C}_t$ to $\mathcal{C}_{t+\Delta t}$ and finally obtains the holo protein-ligand structure at $t = 1$. In the illustration, the gray dots indicate protein residues and the other dots indicate ligand atoms with different element types.

field $u_t(x|x_1)$ and use Conditional Flow Matching (CFM) objective as:

$$\mathcal{L}_{CFM}(\theta) = \mathbb{E}_{t,p_1(x_1),p_t(x|x_1)} \|v_\theta(x,t) - u_t(x|x_1)\|_g^2, \tag{1}$$

because $\mathcal{L}_{FM}$ and $\mathcal{L}_{CFM}$ have the same gradients according to previous works [14, 49]. $t$ is sampled from the uniform distribution between 0 and 1. Once the gradient field $v_\theta$ is learned, we can integrate ordinary differential equations (ODE): $\frac{d}{dt}\phi_t(x) = v_\theta(\phi_t(x), t)$ with $\phi_0(x) = x, \phi_t(x) = x_t$ and ODE solvers [10] to push data from prior distribution $p_0$ to the data distribution $p_1$. Specifically, a data point from the prior distribution $p_0$ is $\mathcal{C}_0 = \{\mathcal{P}_0, \mathcal{G}_0\}$, where $\mathcal{P}_0$ is the initialized protein structure (e.g., the apo conformation) and $\mathcal{G}_0$ is the initialized ligand molecule. The number of ligand atoms is sampled from the reference dataset distribution; the atom types are initialized with uniform distributions; the atom coordinates are initialized with Gaussian distributions inside the protein pocket following [28]. The target data distribution $p_1$ is the holo protein-ligand complex $\mathcal{C}_1$ from datasets.

## 4 FlexSBDD

Figure. 2 shows the overview of FlexSBDD. Given the initialized target protein structure, the goal of FlexSBDD is to generate the binding ligand molecule as well as the adjusted protein structure. In other words, the output is the generated protein-ligand complex. Given the complexity of the protein-ligand system, we first introduce flow matching on the protein backbone (Sec. 4.1), side chain (Sec. 4.2), and ligand atom type (Sec. 4.3) respectively. Then we show E(3)-equivariant network in Sec. 4.4. Finally, we discuss the training and generation procedure of FlexSBDD (Sec. 4.5).

### 4.1 FlexSBDD on Protein Backbone and Ligand Coordinates

Following previous works [9, 41, 77], The backbone atom positions of each residue in a protein can be modeled as a rigid frame $T = (\boldsymbol{x}, \boldsymbol{O}) \in SE(3)$, consisting of $C_\alpha$ coordinate $\boldsymbol{x} \in \mathbb{R}^3$ (we use $\boldsymbol{x}$ to denote coordinates in the rest of this paper) and the frame orientation matrix $\boldsymbol{O} \in SO(3)$. For simplicity, the following deduction focuses on a single residue/frame and can be generalized to all the residues in the protein. The conditional flow of frame $T_t$ is defined to be along the geodesic path connecting $T_0$ (apo frame) and $T_1$ (holo frame): $T_t = \exp_{T_0}(t \log_{T_0}(T_1))$, where $\exp_T$ represents the exponential map and $\log_T$ denotes the logarithmic map at $T$ [77, 9]. Specifically, the conditional flow for the Euclidean coordinate vector $\boldsymbol{x}_t$ and the orientation matrix $\boldsymbol{O}_t$ are defined as:

$$\text{Coordinates}(\mathbb{R}^3)\colon \boldsymbol{x}_t = (1-t)\boldsymbol{x}_0 + t\boldsymbol{x}_1 \tag{2}$$

$$\text{Orientations}(\text{SO}(3))\colon \boldsymbol{O}_t = \exp_{\boldsymbol{O}_0}(t \log_{\boldsymbol{O}_0}(\boldsymbol{O}_1)). \tag{3}$$

Both $\mathbb{R}^3$ and $SO(3)$ are simple manifolds and their closed-form geodesics can be derived. The exponential map $\exp_{\boldsymbol{O}_0}$ can be computed using Rodrigues' formula and the logarithmic map $\log_{\boldsymbol{O}_0}$ is similarly easy to compute with its Lie algebra $\mathfrak{so}(3)$ [77]. In protein-ligand complex, the ligand atom coordinates have the same data modality and probability path as $C_\alpha$. The loss function of FlexSBDD

for protein backbone and ligand coordinates is the summation of the following two terms:

$$\mathcal{L}_{coord}(\theta) = \mathbb{E}_{t,p_1(\boldsymbol{x}_1),p_0(\boldsymbol{x}_0),p_t(\boldsymbol{x}_t|\boldsymbol{x}_0,\boldsymbol{x}_1)} \frac{1}{N_p + N_l} \sum_{i=1}^{N_p+N_l} \left\| v_\theta^{(i)}(\boldsymbol{x}_t^{(i)}, t) - \boldsymbol{x}_1^{(i)} + \boldsymbol{x}_0^{(i)} \right\|_2^2, \quad (4)$$

$$\mathcal{L}_{ori}(\theta) = \mathbb{E}_{t,p_1(\boldsymbol{O}_1),p_0(\boldsymbol{O}_0),p_t(\boldsymbol{O}_t|\boldsymbol{O}_0,\boldsymbol{O}_1)} \frac{1}{N_p} \sum_{i=1}^{N_p} \left\| v_\theta^{(i)}(\boldsymbol{O}_t^{(i)}, t) - \frac{\log_{\boldsymbol{O}_t^{(i)}}(\boldsymbol{O}_1^{(i)})}{1-t} \right\|_{\mathrm{SO}(3)}^2, \quad (5)$$

where the superscript $(i)$ in $\boldsymbol{x}^{(i)}$ and $\boldsymbol{O}^{(i)}$ is the index of the residues/ligand atoms. For conciseness, we use Equ. 4 to represent the coordinate loss with respect to $N_p$ $\alpha$-Carbon and $N_l$ ligand atoms.

## 4.2 FlexSBDD on Sidechain Torsion Angles

In FlexSBDD, we define sidechain torsion angles on the torus space to model the protein sidechain structures. There are maximally four sidechain dihedral angels for each residue i.e., $\boldsymbol{\chi}^{(i)} = \{\chi^{i1}, \chi^{i2}, \chi^{i3}, \chi^{i4}\} \in [0, 2\pi)^4$ and there are totally $4N_p$ torsion angles [81]. Since each torsion angle lies in $[0, 2\pi)$, the $4N_p$ torsion angles of sidechains define a hypertorus $\mathbb{T}^{4N_p}$. The manifold of the hypertorus is parameterized as the quotient space $\mathbb{R}^{4N_p}/2\pi\mathbb{Z}^{4N_p}$, leading to the equivalence relations $\boldsymbol{\chi} = (\chi^{(1)}, \dots, \chi^{(4N_p)}) \sim (\chi^{(1)} + 2\pi, \dots, \chi^{(4N_p)}) \sim (\chi^{(1)}, \dots, \chi^{(4N_p)} + 2\pi)$ [39, 81]. We use the linear interpolation paths and the conditional flow for $\boldsymbol{\chi}$ is defined as:

$$\boldsymbol{\chi}_t = (1-t)\boldsymbol{\chi}_0 + t \cdot \mathrm{reg}(\boldsymbol{\chi}_1 - \boldsymbol{\chi}_0), \quad (6)$$

where $\mathrm{reg}(\cdot)$ means regularizing the torsion angles by $\mathrm{reg}(\boldsymbol{\chi}) = (\boldsymbol{\chi} + \pi) \mod (2\pi) - \pi$. This leads to the closed-from expression of the loss to train the conditional Torus Flow Matching:

$$\mathcal{L}_{sc}(\theta) = \mathbb{E}_{t,p_1(\boldsymbol{\chi}_1),p_0(\boldsymbol{\chi}_0),p_t(\boldsymbol{\chi}_t|\boldsymbol{\chi}_0,\boldsymbol{\chi}_1)} \frac{1}{N_p} \sum_{i=1}^{N_p} \left\| v_\theta^{(i)}(\boldsymbol{\chi}_t^{(i)}, t) - \mathrm{reg}(\boldsymbol{\chi}_1^{(i)} - \boldsymbol{\chi}_0^{(i)}) \right\|_2^2. \quad (7)$$

## 4.3 FlexSBDD on Ligand Atom Types

The ligand atom types are denoted as $\boldsymbol{a} = \{\boldsymbol{a}^{(i)}\}_{i=1}^{N_p}$ where $\boldsymbol{a}^{(i)}$ is the $i$-th atom probability vector with $n_a$ dimensions: $\boldsymbol{a}^{(i)} \in \mathbb{R}^{n_a}$. To build a path, we define $\boldsymbol{a}_0$ as a uniform distribution over all atom types and $\boldsymbol{a}_1$ as the one-hot vector indicating the ground truth atom type. The probability path is define as $\boldsymbol{a}_t = t\boldsymbol{a}_1 + (1-t)\boldsymbol{a}_0$, and $u_t(\boldsymbol{a}|\boldsymbol{a}_0, \boldsymbol{a}_1) = \boldsymbol{a}_1 - \boldsymbol{a}_0$. $\boldsymbol{a}_t$ is a probability vector because its summation over all types equals 1. Following [47, 70, 12], we use Cross-Entropy loss $\mathrm{CE}(\cdot, \cdot)$ to directly measure the difference between the ground truth type and the predicted one:

$$\mathcal{L}_{atom}(\theta) = \mathbb{E}_{t\sim\mathcal{U}(0,1),p_1(\boldsymbol{a}_1),p_0(\boldsymbol{a}_0),p_t(\boldsymbol{a}|\boldsymbol{a}_0,\boldsymbol{a}_1)} \frac{1}{N_l} \sum_{n=1}^{N_l} \mathrm{CE}\left(\boldsymbol{a}_t^{(i)} + (1-t)v_\theta^{(i)}(\boldsymbol{a}_t^{(i)}, t), \boldsymbol{a}_1^{(i)}\right), \quad (8)$$

We also note the recent progress of sequential flow matching methods [71, 12], which can be seamlessly integrated into FlexSBDD and are left for future works.

## 4.4 Model Architecture

FlexSBDD is parameterized with an E(3)-Equivariant Neural Network with scalar-vector dual feature representation to effectively capture the 3D geometric attributes [40, 57]. The scalar features contain basic biochemical knowledge (e.g., residue/atom types), and the vector features contain geometric knowledge of the structure (e.g., direction to the geometric center). The basic building blocks include geometric vector linear (GVL) and geometric vector perceptron (GVP). We also incorporate the geometric vector normalization (GVNorm) and the geometric vector gate (GVGate) for the model's stability and better performance. There are mainly two modules: an encoder that is responsible for encoding the protein-ligand complex 3D graph (see details in Sec. D.2) and a decoder that updates both the coordinates, frame orientation, atom types, and side-chain torsion angles (see details in Sec. D.3). Similar to previous works [57], the update process satisfies the E(3)-equivariance. More model details are in the Appendix. D.5.

## 4.5 Training and Generation

**Training with Data Augmentation:** For protein-ligand complexes from the training set, we associate them with apo conformations from Apobind [3] to create apo-holo pairs. Additionally, we create synthetic apo conformations as data augmentation. This is done by first removing the ligands from the holo proteins, followed by applying OpenMM [21] relaxation and Rosetta repacking [18] to these proteins. For each holo-structure, we generate a total of 9 additional structures: 3 only with sidechain repacking, 3 with both structure relaxation and repacking, and 3 with additional random perturbations with up to 30 degrees to the sidechain angles. In each training iteration, we randomly sample from the corresponding pool of apo structure $\mathcal{C}_0$ of the holo-structure $\mathcal{C}_1$ and interpolate to obtain $\mathcal{C}_t$. The overall loss function is the weighted summation of the above four loss functions:

$$\mathcal{L} = w_{\text{atom}}\mathcal{L}_{\text{atom}} + w_{\text{coord}}\mathcal{L}_{\text{coord}} + w_{\text{ori}}\mathcal{L}_{\text{ori}} + w_{\text{sc}}\mathcal{L}_{\text{sc}}, \tag{9}$$

where $w_{\text{atom}}, w_{\text{coord}}, w_{\text{ori}}$, and $w_{\text{sc}}$ are the loss weights and are set to 2.0, 1.0, 1.0, and 1.0 in the default setting. We adopt Adam [42] optimizer for the optimization and finish training on a Tesla A100 GPU.

**Generation:** Starting with the apo structure and an initialized ligand molecule, denoted as $\mathcal{C}_0$, the generation process of FlexSBDD is the integration of the ODE $\frac{d\mathcal{C}_t}{dt} = v_\theta(\mathcal{C}_t, t)$ from $t = 0$ to $t = 1$ with an Euler solver [10]. Specifically, for each component in $\mathcal{C}_t$, we have:

$$\boldsymbol{x}_{t+\Delta t}^{(i)} = \boldsymbol{x}_t^{(i)} + v_\theta(\boldsymbol{x}_t^{(i)}, t)\Delta t; \quad \boldsymbol{O}_{t+\Delta t}^{(i)} = \boldsymbol{O}_t^{(i)} \exp\left(v_\theta(\boldsymbol{O}_t^{(i)}, t)\Delta t\right); \tag{10}$$

$$\boldsymbol{\chi}_{t+\Delta t}^{(i)} = \text{reg}\left(\boldsymbol{\chi}_t^{(i)} + v_\theta(\boldsymbol{\chi}_t^{(i)}, t)\Delta t\right); \quad \boldsymbol{a}_{t+\Delta t}^{(i)} = \text{norm}\left(\boldsymbol{a}_t^{(i)} + v_\theta(\boldsymbol{a}_t^{(i)}, t)\Delta t\right); \tag{11}$$

where $\Delta t$ is the time step. $\text{norm}(\cdot)$ means normalizing the vector to a probability vector such that its summation is 1, and $\text{reg}(\cdot)$ means regularizing the angles by $\text{reg}(\tau) = (\tau + \pi) \mod (2\pi) - \pi$.

## 5 Experiments

### 5.1 Experimental Setup

**Datasets:** Following previous works [65, 57], we use two popular benchmark datasets for experimental evaluations: CrossDocked and Binding MOAD. **Binding MOAD** dataset [34] contains around 41k experimentally determined protein-ligand complexes. We further filter and split the Binding MOAD dataset based on the proteins' enzyme commission number [7], resulting in 40k protein-ligand pairs for training, 100 pairs for validation, and 100 pairs for testing following previous work [65]. **CrossDocked** dataset [25] contains 22.5 million protein-molecule pairs generated through cross-docking. We use the same data preprocessing and splitting as [52], only high-quality docking poses (RMSD between the docked pose and the ground truth $< 1\text{Å}$) are kept and $30\%$ sequence identity dataset split is adopted. This produces $100,000$ protein-ligand pairs for training and 100 proteins for testing. We regard the protein-ligand structures in the datasets as holo-structures. The corresponding apo structures are obtained from Apobind and the generated apo structures as described in Sec. 4.5. We note that it is fair to compare FlexSBDD with other baseline methods as the additional apo structures contain no ligand molecules and cannot be used by baselines for training.

**Baseline Methods:** We compare FlexSBDD with five representative methods for SBDD. **LiGAN** [62] is a conditional VAE model that represents protein-ligand complex as an atomic density grid. **AR** [52] and **Pocket2Mol** [57] are autoregressive schemes that generate 3D ligand molecules atom-by-atom. **TargetDiff** [28] and **DecompDiff** [29] are state-of-the-art diffusion-based models.

**Evaluation:** We comprehensively evaluate the generated molecules from three perspectives: **binding affinity and molecular properties**, **molecular structures**, and **protein-ligand interactions**: (1) Following previous work [28, 29], we use AutoDock Vina [22] to calculate and report the mean and median of binding affinity-related metrics, including **Vina Score**, **Vina Min**, **Vina Dock**, and **High Affinity**. Vina Score directly measures the binding affinity based on the generated 3D molecules; Vina Min performs a local structure relaxation before calculation; Vina Dock includes an extra step of re-docking, which serves to reveal the optimal binding affinity achievable; High affinity measures the percentage of generated molecules with higher binding affinity than the reference molecule in the

| Methods | Vina Score (↓) | | Vina Min (↓) | | Vina Dock (↓) | | High Affinity (↑) | | QED (↑) | | SA (↑) | | Diversity (↑) | |
|---|---|---|---|---|---|---|---|---|---|---|---|---|---|---|
| | Avg. | Med. | Avg. | Med. | Avg. | Med. | Avg. | Med. | Avg. | Med. | Avg. | Med. | Avg. | Med. |
| Reference | -6.36 | -6.46 | -6.71 | -6.49 | -7.45 | -7.26 | - | - | 0.48 | 0.47 | 0.73 | 0.74 | - | - |
| LiGAN | - | - | - | - | -6.33 | -6.20 | 21.1% | 11.1% | 0.39 | 0.39 | 0.59 | 0.57 | 0.66 | 0.67 |
| AR | -5.75 | -5.64 | -6.18 | -5.88 | -6.75 | -6.62 | 37.9% | 31.0% | 0.51 | 0.50 | 0.63 | 0.63 | 0.70 | 0.70 |
| Pocket2Mol | -5.14 | -4.70 | -6.42 | -5.82 | -7.15 | -6.79 | 48.4% | 51.0% | 0.56 | 0.57 | **0.74** | **0.75** | 0.69 | 0.71 |
| TargetDiff | -5.47 | -6.30 | -6.64 | -6.86 | -7.80 | -7.91 | 58.1% | 59.1% | 0.48 | 0.48 | 0.58 | 0.58 | 0.72 | 0.71 |
| DecompDiff | -5.67 | -6.04 | -7.04 | -6.91 | -8.39 | -8.43 | 64.4% | 71.0% | 0.45 | 0.43 | 0.61 | 0.60 | 0.68 | 0.68 |
| FlexSBDD | **-6.64** | **-7.25** | **-8.27** | **-8.46** | **-9.12** | **-9.25** | 78.5% | 84.2% | **0.58** | **0.59** | 0.69 | 0.73 | **0.76** | 0.75 |

Table 1: Overview of properties of the reference dataset and the molecules generated by different methods on the **CrossDocked** dataset. (↑) / (↓) denotes the larger/smaller, the better. The best results are marked with **bold** and the runner-up with underline.

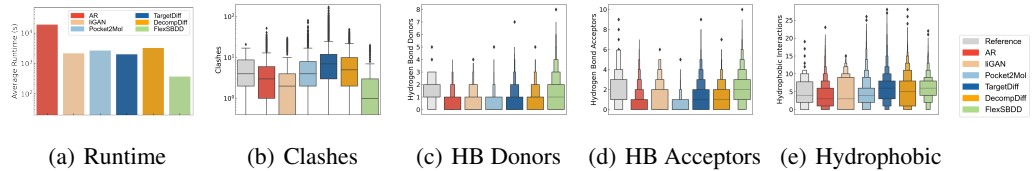

(a) Runtime      (b) Clashes      (c) HB Donors      (d) HB Acceptors      (e) Hydrophobic

Figure 3: Computational Efficiency and Interaction Analysis on CrossDocked. (a) The average time required by different methods to generate 100 ligand molecules for a protein target. (b) The number of steric clashes. (c) The number of hydrogen bond donors in the ligand molecules. (d) The number of hydrogen bond acceptors in the ligand molecules. (e) The number of hydrophobic interactions.

test set. As for the other molecular properties, we consider **QED**, **SA**, and **Diversity**. QED measures how likely a molecule is a potential drug candidate; SA (synthesize accessibility) represents the difficulty of drug synthesis; Diversity is computed as the average pairwise dissimilarity between all generated molecules for a binding pocket. (2) In terms of molecular structures, we calculate the Jensen-Shannon divergences (JSD) in bond length/angle distributions between the reference molecules and the generated molecules following [29]. (3) We adopt PoseCheck [30] to evaluate whether methods can establish favorable interaction between protein and ligand. The interactions include **Hydrogen bonds** and **Hydrophobic interactions**. We also conduct **Steric clashes** analysis to examine unphysical structures/interactions.

## 5.2 Main Results

In Table. 1 and 4, we show the binding affinity and the drug-related properties of the generated molecules on two benchmarks. We can observe that our FlexSBDD outperforms baselines by a large margin in affinity-related metrics. For example, FlexSBDD surpasses the strongest baseline DecompDiff by 0.73 and 0.82 in Avg. and Med. Vina Dock on CrossDokecd and 0.92 and 1.03 respectively on Binding MOAD. These gains indicate the strong capability of FlexSBDD to explore high-affinity drug molecules and adjust protein structures for tight binding. As for molecular properties QED and SA, FlexSBDD also achieves competitive performance with baselines. As discussed in [29], these properties are usually employed as preliminary screening criteria in real drug discovery scenarios as long as they fall into a reasonable range. Finally, the high diversity indicates that FlexSBDD can explore larger chemical space with flexible protein modeling, which is important for early drug discovery. Generation efficiency is also a key factor to consider when sampling a large batch of molecules for screening. A major drawback of widely-used diffusion models is their inference speed, which may require 1000 time steps to produce high-quality samples. In contrast, flow matching methods remove stochasticity from the sampling path and can achieve stable and high-quality generation with much fewer steps (e.g., 20 steps in FlexSBDD). In Figure. 3, we observe that FlexSBDD can generate molecules much more efficiently than autoregressive-based methods such as AR [52] and diffusion-based methods such as TargetDiff [28] and DecompDiff [29].

We further consider steric clashes, hydrogen bonds, and hydrophobic interactions. **Steric Clashes** happens when two neutral atoms come into closer proximity than the combined extent of their van der Waals radii [63], indicating energetically unfavorable and physically unrealistic structures. **Hydrogen bonds (HBs)** [58] and **Hydrophobic interactions** are polar interactions that significantly contribute to the binding affinity between proteins and ligands (More details in Appendix A.1). In Figure. 3, we show the average number of steric clashes, hydrogen bond donors, acceptors, and hydrophobic

interactions in the generated ligands (without redocking). We observe that FlexSBDD can generate ligands introducing fewer clashes and more favorable interactions. For example, the average steric clashes for DecompDiff and FlexSBDD are 6.43 and 1.39 respectively. The average number of HB Acceptors for DecompDiff and FlexSBDD are 1.18 and 1.96 respectively. This could be attributed to the flexible protein adjustment capability of FlexSBDD, which could adaptively adjust protein and ligand conformations to reduce clashes and increase favorable protein-ligand interactions.

In Figure. 4, we show examples of the generated ligand molecules for target proteins. Especially, We colored the original holo protein structure green and the updated structure with FlexSBDD cyan for comparison. Firstly, we observe FlexSBDD can generate ligand molecules with higher affinity and comparable QED and SA compared with reference complexes from datasets and molecules generated by DecompDiff. Moreover, the protein structures of FlexSBDD are adjusted to accommodate the generated ligand molecules. Consistent with the prior knowledge in biology [8], we generally observe that the loop regions in protein structures exhibit greater flexibility, whereas the alpha-helix and beta-sheet regions display more rigidity. To further evaluate the validity of the updated protein structure, we employ self-consistency Template Modeling (scTM) following [48] (more details in Appendix A.2). scTM score ranges from 0 to 1 and a larger scTM score indicates better structural validity. On average, the updated protein structures by FlexSBDD have a scTM score of 0.964, comparable to the score of the original structures from the datasets (0.975). Moreover, to evaluate the validity of sidechain structure, we compute the Mean Absolute Error (MAE) of sidechain angles following previous works [81]. In Table. 6, we observe FlexSBDD achieves better performance in sidechain structure prediction. These results indicate FlexSBDD has learned the protein flexible changes and maintains the structural validity. More results are included in the Appendix B.

## 5.3 Sub-structure analysis

We further conduct sub-structure analysis to evaluate whether FlexSBDD can generate valid molecular conformations. In Table. 2 and Table. 5 in Appendix B, we compute different bond distance and bond angle distributions of the generated molecules and compare them against the corresponding reference empirical distributions following [28, 29]. We can observe that our model has a comparable or better performance on all the bond distances and angles, demonstrating the strong capability of FlexSBDD to generate realistic 3D molecules directly.

| Bond | liGAN | AR | Pocket2 Mol | Target Diff | Decomp Diff | Flex SBDD |
|---|---|---|---|---|---|---|
| C−C | 0.601 | 0.609 | 0.496 | 0.369 | **0.359** | 0.367 |
| C=C | 0.665 | 0.620 | 0.561 | 0.505 | 0.537 | **0.280** |
| C−N | 0.634 | 0.474 | 0.416 | 0.363 | 0.344 | **0.277** |
| C=N | 0.749 | 0.635 | 0.629 | 0.550 | 0.584 | **0.384** |
| C−O | 0.656 | 0.492 | 0.454 | 0.421 | 0.376 | **0.253** |
| C=O | 0.661 | 0.558 | 0.516 | 0.461 | 0.374 | **0.245** |
| C:C | 0.497 | 0.451 | 0.416 | 0.263 | 0.251 | **0.193** |
| C:N | 0.638 | 0.552 | 0.487 | 0.235 | 0.269 | **0.189** |

Table 2: Jensen-Shannon divergence between bond distance distributions of reference and generated ligands (the lower, the better). "-", "=", and ":" denote single, double, and aromatic bonds.

## 5.4 Rediscover Cryptic Pockets with FlexSBDD

The dynamic nature of proteins frequently results in the formation of *cryptic pockets*, which can reveal novel druggable sites not found in static structures and make previously "undruggable" proteins into potential drug targets [55]. To study the capability of FlexSBDD to explore cryptic pockets, we take KRAS$^{G12C}$ for a case study, which is a promising target in the treatment of solid tumors, and over 3 decades of efforts have been devoted to discovering its inhibitors (drug molecules) [17, 32]. The binding mode of ARS-1620 (green, PDB id 5V9U) represents the typical binding pocket exploited by previous research, which limits the exploration of high-affinity inhibitors. Here, we take the protein structure of ARS-1620 as the apo structure and generate ligand

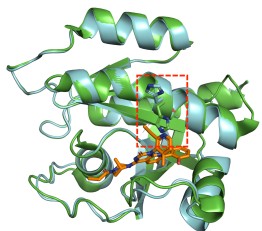

Figure 5: The predicted side chain rotation of residue H95 (marked with the red rectangle) is consistent with experimental observation [45]

molecules. By comparing and filtering the generated molecules according to recent literature [45], we managed to rediscover the cryptic pockets with FlexSBDD. In Figure. 5, the updated structure is colored cyan and the generated ligand molecule is colored orange. We observe that the side chain rotation of residue Histidine-95 (marked with the red rectangle) is consistent with the report in [45] (PDB id 6P8Z), which forms a new subpocket and contributes a lot to binding affinity. This case

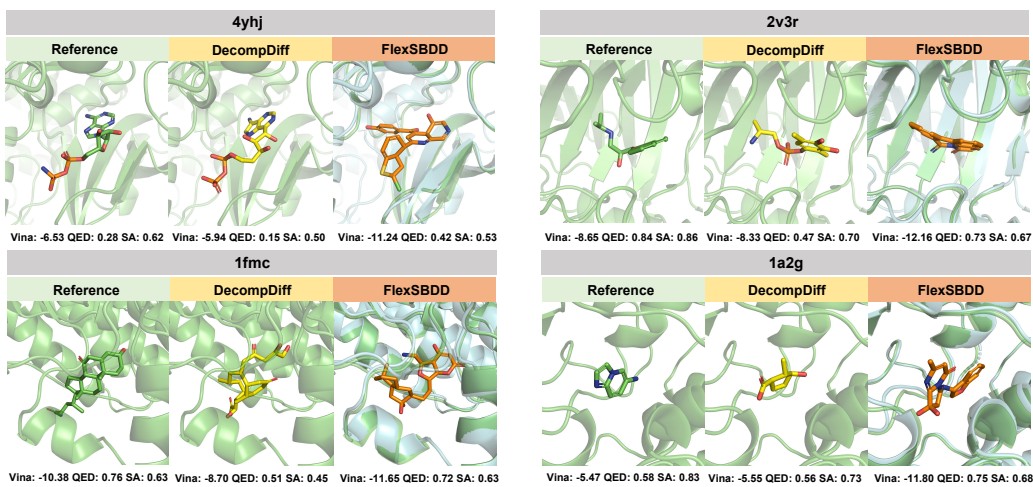

Figure 4: Examples of the generated ligand molecules for target proteins (PDB ID: 4yhj, 2v3r, 1fmc, 1a2g). We colored the original holo-structure from the dataset green and the updated protein structure with FlexSBDD cyan (structures are aligned). The Carbon atoms in Reference, DecompDiff, and FlexSBDD ligands are colored green, yellow, and orange. Vina Score, QED, and SA are reported.

study demonstrates FlexSBDD's capability to accurately model flexible protein structure, update sidechains to reduce steric clashes, and explore cryptic pockets for drug discovery.

## 5.5 Ablation Studies

We conduct a series of ablation experiments to study the effect of different modules on the generation capability of FlexSBDD: (1) **Exp0**: In model training, we remove the data augmentation mentioned in Sec. 5.1, (2) **Exp1**: we replace the geometric vector modules with EGNN [64] adopted in [65], which only has scalar features without vector features, (3) **Exp2**: we do not update the backbone structure of the protein (i.e., $x, O$) (4) **Exp3**: we do not update the sidechain dihedral angles of the protein (i.e., $\chi$), (5) **Exp4**: we fix the whole protein structure in ligand molecule generation. We retrain all the FlexSBDD variant models for comparison. The results are present in Table. 3.

By comparing results from Exp0 and FlexSBDD, we can find that data augmentation indeed helps boost performance by introducing more diverse apo structures. In comparing Exp1 with FlexSBDD, it is obvious that scalar-vector dual feature representation can benefit ligand molecule generation by well capturing geometrical features. When comparing EXP2, 3, and 4 with FlexSBDD, we observe that the modeling of flexible protein structures including backbone and sidechains is important

| Methods | Vina Score (↓) | | Vina Min (↓) | | Vina Dock (↓) | | QED (↑) | |
|---|---|---|---|---|---|---|---|---|
| | Avg. | Med. | Avg. | Med. | Avg. | Med. | Avg. | Med. |
| Exp0 | -6.12 | -6.50 | -7.69 | -7.83 | -8.76 | -8.98 | 0.54 | 0.56 |
| Exp1 | -6.50 | -6.85 | -7.91 | -8.10 | -8.95 | -9.03 | 0.55 | 0.58 |
| Exp2 | -6.57 | -7.19 | -8.14 | -8.31 | -9.05 | -9.20 | 0.56 | 0.57 |
| Exp3 | -6.45 | -7.03 | -7.94 | -8.07 | -8.86 | -8.92 | 0.56 | 0.55 |
| Exp4 | -6.32 | -6.88 | -7.80 | -7.91 | -8.82 | -8.85 | 0.57 | 0.55 |
| FlexSBDD | **-6.64** | **-7.25** | **-8.27** | **-8.46** | **-9.12** | **-9.25** | **0.58** | **0.59** |

Table 3: Effect of different modules on the generation performance of FlexSBDD. The best results are marked with **bold** and the runner-up with underline. The original FlexSBDD is incorporated for comparison.

for FlexSBDD. Specifically, we find modeling the flexibility of sidechain angles is more important than backbone structure as the backbone is more rigid. For example, the average Vina Dock drops to -8.86 for Exp3 (FlexSBDD w/o flexible sidechain) while only drops to -9.05 for Exp2 (FlexSBDD w/o flexible backbone). According to [43], the sidechains are critical to "induced fit", where they adjust positions to accommodate the ligand and enhance binding affinity. Overall, the FlexSBDD variants still demonstrate competitive performance, showing the advantage of flow-matching architecture.

## 5.6 Hyperparameter Analysis

We investigate the influence of two important hyperparameters on the performance of FlexSBDD, the hidden dimension size and the total number of iteration steps $T$ in flow matching. In Figure. 6, we

observe the trend of generating higher-quality molecules with larger hidden dimension sizes and more iteration steps. In the default setting, we set the node scaler feature size to 256 and total iteration steps to 20 to achieve a balance between the computational complexity and the generation quality.

## 6 Conclusion

In this paper, we propose FlexSBDD, a deep generative model capable of modeling the flexible protein structure for ligand molecule generation. FlexSBDD adopts a flow matching framework for efficient ligand generation and leverages E(3)-equivariant network with scalar-vector dual feature representation to effectively model dynamic structural changes. Extensive experiments show its state-of-the-art performance in generating high-affinity molecules with less steric clashes and more favorable interactions. Potential future works include leveraging FlexSBDD to discover more cryptic pockets and modeling other functional proteins such as antibodies, peptides, and enzymes.

## 7 Acknowledgements

This research was supported by grants from the National Natural Science Foundation of China (Grant No. 623B2095) and the Fundamental Research Funds for the Central Universities.

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

# A  Data Analysis

## A.1  Protein-ligand Interaction Analysis

We consider steric clashes, hydrogen bonds, and hydrophobic interactions in protein-ligand interaction analysis with PoseCheck [30]. **Steric Clashes** happens when two neutral atoms come into closer proximity than the combined extent of their van der Waals radii [63], indicating energetically unfavorable and physically unrealistic structures. In PoseCheck, a clash is counted when the pairwise distance between a protein and ligand atom falls below the sum of their van der Waals radii, allowing a clash tolerance of 0.5 Å. **Hydrogen bonds (HBs)** represent a form of molecular interaction where a hydrogen atom, covalently linked to an element of high electronegativity like nitrogen, oxygen, or fluorine, engages with another electronegative atom [58]. These bonds are crucial in numerous protein-ligand interactions [13] and necessitate precise geometric alignments to form [11]. HBs are directional, bestowing distinct roles on the atoms involved: the hydrogen covalently bonded to the electronegative atom acts as a "donor", while the atom that receives the HB is known as an "acceptor". **Hydrophobic interactions** are a type of non-covalent bonding that occurs among hydrophobic molecules or moieties within an aqueous setting. Driven by water's propensity to hydrogen bond with itself, these interactions result in the segregation of non-polar entities, compelling them to cluster together away from the water-rich environment [56]. This behavior significantly contributes to the binding affinity between proteins and ligands.

## A.2  Protein Sturcture Analysis

Following [48], scTM takes a generated structure and feeds it into ProteinMPNN [19], a state-of-the-art structure-conditioned sequence generation method. With a sampling temperature of 0.1, we generate eight sequences per input structure and then use OmegaFold [75] to predict the structure of each putative sequence. We follow [48] that substitutes AlphaFold2 with OmegaFold for better sequence prediction performance. Finally, scTM is measured by computing the TM-score [80], a metric of structural congruence of the OmegaFold-predicted structure and the original updated structure by our FlexSBDD. scTM scores range from 0 to 1, with higher numbers corresponding to the increased likelihood that an input structure is designable (higher structural validity).

# B  More Results and Analysis

Table. 5, 7 and Figure. 6 show the additional results on substructure analysis, and hyperparameters analysis.

## B.1  Benchmark Results on CrossDocked

In Table. 4, we show the additional results on the Binding MOAD dataset.

| Methods | Vina Score (↓) | | Vina Min (↓) | | Vina Dock (↓) | | High Affinity (↑) | | QED (↑) | | SA (↑) | | Diversity (↑) | |
|---|---|---|---|---|---|---|---|---|---|---|---|---|---|---|
| | Avg. | Med. | Avg. | Med. | Avg. | Med. | Avg. | Med. | Avg. | Med. | Avg. | Med. | Avg. | Med. |
| Reference | -6.68 | -6.66 | -7.62 | -7.47 | -8.21 | -8.16 | - | - | 0.60 | 0.58 | 0.33 | 0.34 | - | - |
| LiGAN | - | - | - | - | -7.09 | -6.96 | 18.9% | 16.2% | 0.37 | 0.38 | 0.37 | 0.40 | 0.68 | 0.69 |
| AR | -6.19 | -6.04 | -6.86 | -6.80 | -7.65 | -7.61 | 32.6% | 31.9% | 0.42 | 0.40 | 0.35 | 0.36 | 0.68 | 0.69 |
| Pocket2Mol | -6.02 | -5.97 | -6.71 | -6.80 | -7.69 | -7.74 | 36.9% | 37.1% | 0.60 | 0.59 | 0.34 | 0.35 | 0.70 | 0.72 |
| TargetDiff | -6.13 | -6.20 | -6.85 | -6.92 | -7.95 | -7.94 | 40.3% | 39.7% | 0.50 | 0.48 | 0.29 | 0.33 | 0.71 | 0.70 |
| DecompDiff | -6.37 | -6.41 | -7.52 | -7.31 | -8.46 | -8.51 | 56.4% | 58.3% | 0.60 | 0.61 | 0.32 | 0.34 | 0.68 | 0.66 |
| FlexSBDD | **-7.04** | **-7.20** | **-8.36** | **-8.73** | **-9.38** | **-9.54** | **74.5%** | **76.9%** | **0.63** | **0.64** | **0.42** | **0.41** | **0.73** | **0.74** |

Table 4: Overview of properties of the reference dataset and the molecules generated by different methods on **Binding MOAD** dataset. (↑) / (↓) denotes the larger/smaller, the better. The best results are marked with **bold** and the runner-up with underline.

## B.2  Bond Angle Distributions

In Table. 5, we show the Jensen-Shannon divergence between bond angle distributions of the reference molecules and the generated molecules.

| Bond | liGAN | AR | Pocket2 Mol | Target Diff | Decomp Diff | Flex SBDD |
|---|---|---|---|---|---|---|
| CCC | 0.598 | 0.340 | 0.323 | 0.328 | 0.314 | **0.285** |
| CCO | 0.637 | 0.442 | 0.401 | 0.385 | 0.324 | **0.316** |
| CNC | 0.604 | 0.419 | 0.237 | 0.367 | 0.297 | **0.226** |
| OPO | 0.512 | 0.367 | 0.274 | 0.303 | 0.217 | **0.210** |
| NCC | 0.621 | 0.392 | 0.351 | 0.354 | 0.294 | **0.283** |
| CC=O | 0.636 | 0.476 | 0.353 | 0.356 | **0.259** | 0.270 |
| COC | 0.606 | 0.459 | **0.317** | 0.389 | 0.339 | 0.320 |

Table 5: Jensen-Shannon divergence between bond angle distributions of the reference molecules and the generated molecules, and lower values indicate better performances. We highlight the best two results with **bold text** and underlined text, respectively.

## B.3 Validity of Side Chain Prediction

In FlexSBDD, the sidechain torsion angles are predicted and the sidechain conformations are derived based on the dihedral angles and the ideal bond length/angles. To evaluate the validity of sidechain structure, we compute the Mean Absolute Error (MAE) of sidechain angles (degrees) following previous works [81] in Table. 6. We also compare the results with NeuralPlexer [61], the state-of-the-art protein-ligand complex structure prediction. In the table, we report the average MAE and can observe that FlexSBDD achieves better performance in generating valid sidechain structures.

| Method | $\chi_1$ | $\chi_2$ | $\chi_3$ | $\chi_4$ |
|---|---|---|---|---|
| NeuralPlexer | 15.40 | 18.77 | 44.83 | 50.24 |
| FlexSBDD | **12.95** | **17.80** | **32.18** | **46.71** |

Table 6: The MAE of NeuralPlexer and FlexSBDD on sidechain torsion angles.

## B.4 More Results on Abation Studies

In Table. 7, we show the average number of interactions in the ablation studies.

| Methods | Steric Clashes ($\downarrow$) | HB Donors ($\uparrow$) | HB Acceptors ($\uparrow$) | Hydrophobic ($\uparrow$) |
|---|---|---|---|---|
| Exp0 | 1.56 | 1.38 | 1.74 | 4.79 |
| Exp1 | 1.77 | 1.35 | 1.77 | 5.40 |
| Exp2 | 1.45 | 1.23 | 1.85 | 5.75 |
| Exp3 | 1.92 | 1.11 | 1.62 | 5.23 |
| Exp4 | 1.90 | 1.09 | 1.58 | 5.29 |
| FlexSBDD | **1.39** | **1.40** | **1.96** | **6.12** |

Table 7: Effect of different modules on the generation performance of FlexSBDD. We show the average number of interactions here. We highlight the best two results with **bold text** and underlined text, respectively.

## B.5 Evaluation of Binding Affinity with GlideSP

Besides Vina Scores, we also try to leverage other docking methods to evaluate the binding affinity [24]. In Table. 8, we further leverage Glide [26] to evaluate the generated ligand molecules, which demonstrates superior ability in filtering active compounds. Specifically, we calculate the min-in-place GlideSP score following [24], where the ligand structure undergoes force-field-based energy minimization within the receptor's field before scoring. In the table below, we observe that FlexSBDD can also achieve the best score on Glide, demonstrating its strong performance in generating protein-binding molecules.

Table 8: Comparison of Average Min-in-Place GlideSP Scores on CrossDocked.

| Methods | Avg. min-in-place GlideSP score (↓) |
|---|---|
| Reference | -6.32 |
| LiGAN | -6.14 |
| AR | -6.20 |
| Pocket2Mol | -6.71 |
| TargetDiff | -6.86 |
| DecompDiff | -7.09 |
| **FlexSBDD** | **-7.55** |

### B.6 Hyperparameter Analysis

In Figure. 6, we show hyperparameter analysis for the hidden dimension size and the total steps of flow matching.

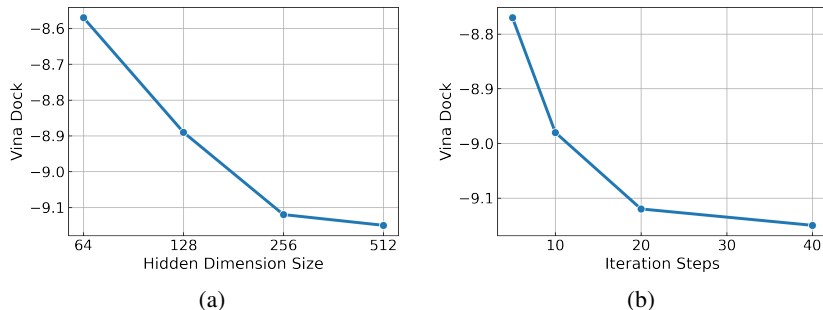

(a)                                      (b)

Figure 6: Hyperparameters Analysis with respect to (a) the hidden dimension size (the node scaler features) and (b) the total iteration steps of flow matching. When varying the dimension size of the node scaler features, the other features are scaled proportionally.

## C  More Details of FlexSBDD Training and Generation

### C.1  Hyperparameters settings

To construct the protein-ligand KNN graph, we set $k$ as 8 (each node is connected to its nearest 8 neighbors). In the default setting, we use a hidden size of 256, 128, 128, and 64 for the scalar features of nodes, scalar features of edges, vector features of nodes, and vector features of edges, respectively. The Encoder and Decoder have 6 layers respectively with the number of attention heads set as 4. The number of integration steps in flow matching is 20 for FlexSBDD. The hyperparameters for the loss function: $w_{\text{atom}}, w_{\text{coord}}, w_{\text{ori}}$, and $w_{\text{sc}}$ are selected based on grid search ($\{0.5, 1.0, 2.0, 3.0\}$). $w_{\text{atom}}, w_{\text{coord}}, w_{\text{ori}}$, and $w_{\text{sc}}$ are set to 2.0, 1.0, 1.0, and 1.0 in the default setting. To train FlexSBDD, we use the Adam [42] as our optimizer with a learning rate of 0.001, $betas = (0.95, 0.999)$, and batch size 4 for 500k iterations. It takes around 36 hours on n one NVIDIA GeForce GTX A100 GPU to complete the training.

### C.2  Pseudo Algorithms

We show the pseudo codes of FlexSBDD training and generation in Algorithm 1 and 2.

## D  More Details of Neural Network Architecture

### D.1  Overview

A series of previous works on biochemistry tasks [40, 57, 79, 20] have shown that representing the nodes and edges in 3D graphs with scalar-vector dual features can greatly improve performance. In

---

**Algorithm 1:** Training algorithm of FlexSBDD

---

**Input:** Holo data distribution $p_1$, FlexSBDD model $v_\theta$, Apobind and generated apo structures
**while** *Training* **do**
    $\mathcal{C}_1 \sim p_1; t \sim \mathcal{U}[0,1]$;
    Sample $\mathcal{C}_0$ from the corresponding apo structure pool (Apobind and data augmentation)
    $\boldsymbol{x}_t = (1-t)\boldsymbol{x}_0 + t\boldsymbol{x}_1;\quad \boldsymbol{O}_t = \exp_{\boldsymbol{O}_0}(t\log_{\boldsymbol{O}_0}(\boldsymbol{O}_1))$
    $\boldsymbol{\chi}_t = (1-t)\boldsymbol{\chi}_0 + t \cdot \text{reg}(\boldsymbol{\chi}_1 - \boldsymbol{\chi}_0);\quad \boldsymbol{a}_t = t\boldsymbol{a}_1 + (1-t)\boldsymbol{a}_0;$ // Interpolation
    $\mathcal{L} \leftarrow w_{\text{atom}}\mathcal{L}_{\text{atom}} + w_{\text{coord}}\mathcal{L}_{\text{coord}} + w_{\text{ori}}\mathcal{L}_{\text{ori}} + w_{\text{sc}}\mathcal{L}_{\text{sc}}$ // calculate loss according to Equ. 9;
    $\theta \leftarrow \textbf{Update}(\theta, \nabla_\theta \mathcal{L}_{FM})$;
**return** $v_\theta$

---

---

**Algorithm 2:** Generation algorithm of FlexSBDD

---

**Input:** Total number of integration steps T, and trained model $v_\theta$
$steps \leftarrow 0, t \leftarrow 0, \Delta t \leftarrow 1/T$;
Initialize $\mathcal{C}_0$ with the apo structure and sampled ligand atoms;
**while** $steps \leq T-1$ **do**
    $\boldsymbol{x}_{t+\Delta t}^{(i)} = \boldsymbol{x}_t^{(i)} + v_\theta(\boldsymbol{x}_t^{(i)}, t)\Delta t;\quad \boldsymbol{O}_{t+\Delta t}^{(i)} = \boldsymbol{O}_t^{(i)}\exp\left(v_\theta(\boldsymbol{O}_t^{(i)}, t)\Delta t\right);$
    $\boldsymbol{\chi}_{t+\Delta t}^{(i)} = \text{reg}(\boldsymbol{\chi}_t^{(i)} + v_\theta(\boldsymbol{\chi}_t^{(i)}, t)\Delta t);\quad \boldsymbol{a}_{t+\Delta t}^{(i)} = \text{norm}\left(\boldsymbol{a}_t^{(i)} + v_\theta(\boldsymbol{a}_t^{(i)}, t)\Delta t\right);$
    $t \leftarrow t + \Delta t$ ;
**return** $\mathcal{C}_1$

---

FlexSBDD, all nodes and edges in the target protein $\mathcal{P}$ and the generated molecules $\mathcal{G}$ are assigned with both scalar and vector features to better capture the 3D geometric information. The scalar features contain basic biochemical knowledge (e.g., residue/atom types), and the vector features contain geometric knowledge of the structure (e.g., direction to the geometric center). In the rest of this paper, we use "·" and "→" overheads to explicitly indicate scalar features and vector features (e.g., $\dot{\mathbf{v}}$ and $\vec{\mathbf{v}}$).

We adopt the geometric vector linear (GVL) and the geometric vector perceptron (GVP) as the main building blocks to enhance the information flows between the scalar features and the vector features and achieve E(3)-equivariance [40]. The details of GVP and GVL are shown in Appendix D.5. Briefly, they propagate the vector features into the scalar features by row-wise norm and propagate the scalar features to the vector features through gating. The GVP further applies extra non-linear transformations to both the scalar and vector features, following the output of GVL:

$$
\begin{aligned}
(\dot{\mathbf{v}}', \vec{\mathbf{v}}') &\leftarrow \text{GVL}(\dot{\mathbf{v}}, \vec{\mathbf{v}}), \\
(\dot{\mathbf{v}}', \vec{\mathbf{v}}') &\leftarrow \text{NonLinearTransform}(\dot{\mathbf{v}}', \vec{\mathbf{v}}'),
\end{aligned}
\tag{12}
$$

where $(\dot{\mathbf{v}}, \vec{\mathbf{v}})$ could be any pair of scalar-vector features. Incorporating vector features is essential for our model's ability to directly and precisely update the positions of atoms and model the conformation change of the flexible protein. In our model, we also incorporate the geometric vector normalization (GVNorm) and the geometric vector gate (GVGate) for model's stability and better performance. Specifically, GVNorm combines the layer normalization [6] with the vector normalization [40]; GVGate performs skip connection and fuses features from different blocks.

### D.2 Encoder

In FlexSBDD, we represent the protein pocket-ligand complex as a $k$-nearest neighbor (KNN) graph in which nodes represent protein residues or ligand atoms and each node is connected to its $k$-nearest neighbors. The input scalar features of residues are onehot embeddings of residue types and the input scalar features of ligand atoms are initialized with a uniform distribution over all atom types. The input vector features of residues are computed based on the coordinates of backbone and sidechain atoms, while the vector features of ligand atoms are the Euclidean vectors pointing to the geometric center of the ligand molecule (see Appendix D.4). The scalar edge features are the radial basis function (RBF) distance encodings [66] and the vector edge features are the relative coordinates. In

the flow matching model, to further incorporate time step information, we embed time with sinusoidal embedding [73] and concatenate it with the input scalar node features following [28].

Generally, the encoder of FlexSBDD follows the message-passing paradigm to update the features. Denote the feature of node $i$ at the $l$-th layer as $(\dot{\mathbf{v}}_i^{(l)}, \vec{\mathbf{v}}_i^{(l)})$ and the edge feature between node $i$ and $j$ as $(\dot{\mathbf{e}}_{ij}^{(l)}, \vec{\mathbf{e}}_{ij}^{(l)})$ (we use subscript here for the index of nodes instead of time step in the main paper). Each layer consists of a message-passing module $M_l$ and an update module $U_l$:

$$\mathbf{M}_v^{(l)}, \mathbf{M}_e^{(l)} = M_l(\dot{\mathbf{v}}_j^{(l-1)}, \vec{\mathbf{v}}_j^{(l-1)}, \dot{\mathbf{e}}_{ij}^{(l-1)}, \vec{\mathbf{e}}_{ij}^{(l-1)}),$$
$$(\dot{\mathbf{v}}_i^{(l)}, \vec{\mathbf{v}}_i^{(l)}), (\dot{\mathbf{e}}_{ij}^{(l)}, \vec{\mathbf{e}}_{ij}^{(l)}) = U_l(\dot{\mathbf{v}}_i^{(l-1)}, \vec{\mathbf{v}}_i^{(l-1)}, \mathbf{M}_v^{(l)}, \mathbf{M}_e^{(l)}),$$

(13)

where we use $\mathbf{M}_v^{(l)} = (\dot{\mathbf{m}}_i^{(l)}, \vec{\mathbf{m}}_i^{(l)})$ and $\mathbf{M}_e^{(l)} = (\dot{\mathbf{m}}_{ij}^{(l)}, \vec{\mathbf{m}}_{ij}^{(l)})$ to denote the calculated messages for node $i$ and the edge between node $i$ and $j$. The message-passing module is based on an attention mechanism [73]. The query $(\dot{\mathbf{q}}_i, \vec{\mathbf{q}}_i)$, key $(\dot{\mathbf{k}}_j, \vec{\mathbf{k}}_j)$, value $(\dot{\mathbf{u}}_j, \vec{\mathbf{u}}_j)$, and edge bias $(\dot{\mathbf{b}}_{ij}, \vec{\mathbf{b}}_{ij})$ are first calculated with GVLs (the layer superscripts are omitted here for simplicity):

$$\dot{\mathbf{q}}_i, \vec{\mathbf{q}}_i = \text{GVL}(\dot{\mathbf{v}}_i^{(l-1)}, \vec{\mathbf{v}}_i^{(l-1)}),$$
$$\dot{\mathbf{k}}_j, \vec{\mathbf{k}}_j, \dot{\mathbf{u}}_j, \vec{\mathbf{u}}_j = \text{GVL}(\dot{\mathbf{v}}_j^{(l-1)}, \vec{\mathbf{v}}_j^{(l-1)}),$$
$$\dot{\mathbf{b}}_{ij}, \vec{\mathbf{b}}_{ij} = \text{GVL}(\dot{\mathbf{e}}_{ij}^{(l-1)}, \vec{\mathbf{e}}_{ij}^{(l-1)}),$$

(14)

Then the attention weights for the scalar are computed as:

$$\dot{\mathbf{a}}_{ij} = \dot{\mathbf{q}}_i \odot \dot{\mathbf{k}}_j \odot \dot{\mathbf{b}}_{ij}, \quad \dot{\mathbf{a}}_{ij} \in \mathbb{R}^{h^s}$$
$$\hat{a}_{ij} = \text{softmax}_j \frac{1}{\sqrt{h^s}} \dot{\mathbf{a}}_{ij} \mathbf{1},$$

(15)

Similarly for the vector features:

$$\vec{\mathbf{a}}_{ij} = \vec{\mathbf{q}}_i \odot \vec{\mathbf{k}}_j \odot \vec{\mathbf{b}}_{ij}, \quad \vec{\mathbf{a}}_{ij} \in \mathbb{R}^{h^v \times 3},$$
$$\hat{\vec{a}}_{ij} = \text{softmax}_j \frac{1}{\sqrt{3h^v}} \mathbf{1}^\top \vec{\mathbf{a}}_{ij} \mathbf{1},$$

(16)

where $\odot$ denotes the Hadamard product, $\mathbf{1}$ is the vector with all entries as 1, $h^s$ and $h^v$ denote the hidden dimension size of the scalar and vector features respectively. $\text{softmax}_j$ means perform softmax over the $j$ index. $\hat{a}_{ij}$ and $\hat{\vec{a}}_{ij}$ are the attention weights (scalar and vector channel) between node $i$ and $j$. The message is obtained as:

$$(\dot{\mathbf{m}}_i^{(l)}, \vec{\mathbf{m}}_i^{(l)}) = \text{GVL}(\sum_j \hat{a}_{ij} \dot{\mathbf{u}}_j, \sum_j \hat{\vec{a}}_{ij} \vec{\mathbf{u}}_j),$$
$$(\dot{\mathbf{m}}_{ij}^{(l)}, \vec{\mathbf{m}}_{ij}^{(l)}) = \text{GVL}(\mathbf{a}_{ij}, \vec{\mathbf{a}}_{ij}),$$

(17)

where the message features with respect to node $i$ are obtained by applying GVL to the weighted summation of the neighboring value features. Finally, the update module $U_l$ is formulated as:

$$\dot{\mathbf{v}}_i^{(l)}, \vec{\mathbf{v}}_i^{(l)} = \text{GVNorm}(\text{GVGate}(\dot{\mathbf{v}}_i^{(l-1)}, \vec{\mathbf{v}}_i^{(l-1)}, \dot{\mathbf{m}}_i^{(l)}, \vec{\mathbf{m}}_i^{(l)})),$$
$$\dot{\mathbf{e}}_{ij}^{(l)}, \vec{\mathbf{e}}_{ij}^{(l)} = \text{GVNorm}(\text{GVGate}(\dot{\mathbf{e}}_{ij}^{(l-1)}, \vec{\mathbf{e}}_{ij}^{(l-1)}, \dot{\mathbf{m}}_{ij}^{(l)}, \vec{\mathbf{m}}_{ij}^{(l)})),$$

(18)

where the GVGate fuses the information from the $(l-1)$-th layer and the calculated messages from the $l$-th layer. GVNorm is appended to normalize the features.

### D.3 Decoder

The node/edge features of the decoder are initialized from the output of the encoder and are updated the same as the encoder. The decoder of FlexSBDD further updates the protein $C_\alpha$ and ligand atom coordinates as follows:

$$\dot{\boldsymbol{f}}_i^{(l)}, \vec{\boldsymbol{f}}_i^{(l)} = \text{GVL}(\text{GVP}(\sum_j \hat{a}_{ij} \dot{\mathbf{u}}_j, \sum_j \hat{\vec{a}}_{ij} \vec{\mathbf{u}}_j)),$$
$$\vec{\boldsymbol{r}}_i^{(l)} = \sum_j \frac{\boldsymbol{x}_i^{(l-1)} - \boldsymbol{x}_j^{(l-1)}}{\|\boldsymbol{x}_i^{(l-1)} - \boldsymbol{x}_j^{(l-1)}\|_2} \text{MLP}(\text{concat}(\dot{\mathbf{a}}_{ij}, \|\vec{\mathbf{a}}_{ij}\|_2^{(r)})),$$
$$\boldsymbol{x}_i^{(l)} = \boldsymbol{x}_i^{(l-1)} + \vec{\boldsymbol{f}}_i^{(l)} + \vec{\boldsymbol{r}}_i^{(l)},$$

(19)

where $\boldsymbol{x}_i^{(l)}$ is the node $i$'s coordinate at the $l$-th layer and $\|\cdot\|_2^{(r)}$ denotes the row-wise L2 norm. The attention weights $(\hat{\hat{a}}_{ij}, \hat{a}_{ij}, \dot{\mathbf{a}}_{ij}, \vec{\mathbf{a}}_{ij})$ and features $(\dot{\mathbf{u}}_j, \vec{\mathbf{u}}_j)$ are calculated similarly to Equations 14 and 15. Finally, we apply MLPs on last layer representations $(\dot{\mathbf{v}}_i^{(L)}, \vec{\mathbf{v}}_i^{(L)})$ (totally $L$ layers) that capture the chemical and geometric attributes for ligand atom type $\boldsymbol{a}^{(i)}$, residue sidechain dihedral angles $\boldsymbol{\chi}^{(i)}$, and the residue backbone orientation $\boldsymbol{O}^{(i)}$ prediction. For the efficient encapsulation of three-dimensional rotations for $\boldsymbol{O}^{(i)}$, we predict a unit quaternion vector [38]. The quaternion can be easily transformed into a rotation matrix and is a more concise representation of a rotation in 3D. The protein-ligand structure is then adjusted based on the updated coordinates, orientation, and the sidechain dihedral angles. Similar to previous works [57], the update process satisfies the E(3)-equivariance.

### D.4    Feature Initialization

**Protein node vector features**    Following [20], the vector feature for protein nodes in FlexSBDD consists of three parts with a total dimension of $[N_p, 24, 3]$ ($N_p$ is the total number of protein nodes).

(1) Euclidean vectors between the C, CA, N, CB (CA for GLY) atoms for a given residue (shape: $[N_p, 16, 3]$). It encompasses various combinations like C to C, C to CA, C to N, C to CB, CA to C, and so forth.

(2) Euclidean vectors between atom $j$ and atom $k$ in for all side-chain dihedral angles (shape: $[N_p, 4, 3]$). For instance, in an amino acid like Arginine (ARG), the side-chain angles (denoted as $\chi_1, \chi_2, \chi_3, \chi_4$) are defined by specific sequences of four atoms ($i$-$j$-$k$-$l$) according to Rosetta: $\chi_1$: N-CA-CB-CG, $\chi_2$: CA-CB-CG-CD, $\chi_3$: CB-CG-CD-NE, $\chi_4$: CG-CD-NE-CZ. Then the Euclidean vectors can be obtained by combining vectors of CA to CB, CB to CG, CG to CD, and CD to NE. For residues with less than 4 sidechain angles, the corresponding vectors are assigned 0.

(3) Euclidean vectors between CA and atom $k$ in all side-chain dihedral angles (shape: $[N_p, 4, 3]$). For example, the vectors for ARG can be obtained by combining Euclidean vectors of CA to CB, CA to CG, CA to CD, and CA to NE. For residues with less than 4 sidechain angles, the corresponding vectors are assigned 0.

**Ligand node vector features**    The vector features for ligand nodes are initialized as the Euclidean vectors between ligand atoms and the geometric center of the ligand molecule (shape: $[N_l, 1, 3]$). $N_l$ is the number of ligand atoms.

### D.5    Geometric Vector Modules

In Algorithm 3 and 4, we show the Geometric vector linear (GVL) and the Geometric vector perception (GVP) modules in FlexSBDD. In Algorithm 5 and 5, we show the GVNorm and GVGate modules for the stability and better performance of FlexSBDD.

---

**Algorithm 3:** Geometric Vector Linear (GVL)

---

**Input:** Scalar and vector features $(\mathbf{v}, \vec{\mathbf{v}})$
**Output:** Updated scalar and vector features $(\mathbf{v}^u, \vec{\mathbf{v}}^u)$
**Function** GVL$(\mathbf{v}, \vec{\mathbf{v}})$**:**
   |   $\vec{\mathbf{v}}' \leftarrow$ LinearNoBias$(\vec{\mathbf{v}})$;
   |   $\mathbf{v}' \leftarrow \|\vec{\mathbf{v}}'\|_2$;
   |   $\mathbf{v}'' \leftarrow$ concat$(\mathbf{v}, \mathbf{v}')$;
   |   $\mathbf{v}^u \leftarrow$ Linear$(\mathbf{v}'')$;
   |   $\vec{\mathbf{v}}'' \leftarrow$ LinearNoBias$(\vec{\mathbf{v}}')$;
   |   $\vec{\mathbf{v}}^u \leftarrow$ sigmoid$(\mathbf{v}^u) \odot \vec{\mathbf{v}}''$;
   |   **return** $(\mathbf{v}^u, \vec{\mathbf{v}}^u)$;

---

**Algorithm 4:** Geometric Vector Perceptron (GVP)

**Input:** Scalar and vector features $(\mathbf{v}, \vec{\mathbf{v}})$
**Output:** Nonlinear transformed scalar and vector features $(\mathbf{v}^u, \vec{\mathbf{v}}^u)$
**Function** GVP($\mathbf{v}, \vec{\mathbf{v}}$)**:**
    $\mathbf{v}', \vec{\mathbf{v}}' \leftarrow \text{GVL}(\mathbf{v}, \vec{\mathbf{v}})$;
    $\mathbf{v}^u \leftarrow \text{leaky\_relu}(\mathbf{v}')$;
    $\vec{\mathbf{v}}'' \leftarrow \text{LinearNoBias}(\vec{\mathbf{v}}')$;
    $\mathbf{v}_{dot} \leftarrow (\vec{\mathbf{v}}' \odot \vec{\mathbf{v}}'')\mathbf{1}$;
    $\mathbf{v}_{mask} \leftarrow 1 \text{ if } \mathbf{v}_{dot} \geq 0, \text{ else } 0$;
    $\vec{\mathbf{v}}_{act} \leftarrow \mathbf{v}_{dot} \oslash \|\vec{\mathbf{v}}''\|_2^2 \odot \vec{\mathbf{v}}$;   // $\oslash$ is element-wise division
    $\vec{\mathbf{v}}^u \leftarrow \alpha\vec{\mathbf{v}}' + (1 - \alpha)\left(\mathbf{v}_{mask} \odot \vec{\mathbf{v}}' + (1 - \mathbf{v}_{mask}) \odot (\vec{\mathbf{v}}' - \vec{\mathbf{v}}_{act})\right)$;   // $\alpha = 0.01$
    **return** $(\mathbf{v}^u, \vec{\mathbf{v}}^u)$;

---

**Algorithm 5:** Geometric Vector Normalization (GVNorm)

**Input:** Scalar and vector features $(\mathbf{v}, \vec{\mathbf{v}})$
**Output:** Normalized scalar and vector features $(\mathbf{v}^u, \vec{\mathbf{v}}^u)$
**Function** GVNorm($\mathbf{v}, \vec{\mathbf{v}}$)**:**
    $\mathbf{v}^u \leftarrow \text{LayerNorm}(\mathbf{v})$;
    $\vec{\mathbf{v}}' \leftarrow \vec{\mathbf{v}}/\sqrt{\frac{1}{h'}\langle\vec{\mathbf{v}}, \vec{\mathbf{v}}\rangle_F}$;   // $\vec{\mathbf{v}}' \in \mathbb{R}^{h' \times 3}$
    $\vec{\mathbf{v}}^u \leftarrow \gamma\vec{\mathbf{v}}' + \beta$;   // $\gamma \in \mathbb{R}^1, \beta \in \mathbb{R}^1$ are trainable parameters
    **return** $(\mathbf{v}^u, \vec{\mathbf{v}}^u)$;

---

**Algorithm 6:** Geometric vector gate (GVGate)

**Input:** Scalar and vector features $(\mathbf{v}, \vec{\mathbf{v}})$
**Output:** Updated scalar and vector features $(\mathbf{v}^u, \vec{\mathbf{v}}^u)$
**Function** GVGate($\mathbf{v}_p, \mathbf{v}_q, \vec{\mathbf{v}}_p, \vec{\mathbf{v}}_q$)**:**
    $\mathbf{v}_c \leftarrow \text{concat}(\mathbf{v}_p, \mathbf{v}_q, \mathbf{v}_p - \mathbf{v}_q)$;
    $\vec{\mathbf{v}}_c \leftarrow \text{concat}(\vec{\mathbf{v}}_p, \vec{\mathbf{v}}_q, \vec{\mathbf{v}}_p - \vec{\mathbf{v}}_q)$;
    $\mathbf{v}_g, \vec{\mathbf{v}}_g \leftarrow \text{GVL}(\mathbf{v}_c, \vec{\mathbf{v}}_c)$;
    $\mathbf{g}_s \leftarrow \text{sigmoid}(\mathbf{v}_g)$;
    $\mathbf{g}_v \leftarrow \text{sigmoid}(\|\vec{\mathbf{v}}_g\|_2)$;
    $\mathbf{v}^u \leftarrow \mathbf{g}_s \odot \mathbf{v}_p + (1 - \mathbf{g}_s) \odot \mathbf{v}_q$;
    $\vec{\mathbf{v}}^u \leftarrow \mathbf{g}_v \odot \vec{\mathbf{v}}_p + (1 - \mathbf{g}_v) \odot \vec{\mathbf{v}}_q$;
    **return** $(\mathbf{v}^u, \vec{\mathbf{v}}^u)$

## E   Limitations and Broader Impact

One limitation of FlexSBDD is that it only considers small molecule design. Recently, other drug modalities such as antibodies, peptides, and nucleic acids have played critical roles in drug discoveries and bio-engineering. We would like to build a generalized version of FlexSBDD for other drug modalities. Another limitation is the limited dataset size, which restricts the scaling of the proposed models. In the future, we may benefit from the generated protein-ligand interaction data from generative AI models e.g., AlphaFold 3 [1] and RoseTTAFold All-Atom [44].

As for the broader impacts, there are many potential applications of our work, e.g., discovering cryptic pockets and generating drugs to cure various diseases. We acknowledge the necessity for regulatory oversight of our Structure-Based Drug Design (SBDD) technique to prevent the creation of harmful molecules. Overall, we believe the positive influence of our work outweighs the potential negative impacts.

