# OpenReview forum: "FlexSBDD: Structure-Based Drug Design with Flexible Protein Modeling"
_NeurIPS.cc/2024/Conference — NeurIPS 2024 poster_

### Official Review · Reviewer_7REV · 2024-07-03

**Soundness:** 3
**Presentation:** 2
**Contribution:** 3
**Rating:** 6
**Confidence:** 3

**Summary:**

This paper presents a new deep generative model, FlexSBDD, which advances the field of structure-based drug design (SBDD) by accounting for the flexibility of proteins when generating 3D ligand molecules. This approach addresses the shortcomings of traditional SBDD methods that assume proteins are rigid, leading to less effective drug interactions. To obtain apo-holo data pairs, this paper utilizes Apobind to generate apo strutures from known holo structures. This paper adopts advanced flow matching to learn the apo-holo dynamics of protein and generate 3D molecules in surprising 20 steps. Experiments on CrossDocked and Bingding MOAD show that FlexSBDD can generate drug-like molecules with highest affinity compared to auto-regressive and diffusion-based baselines.

**Strengths:**

1. For task, this paper focuses on protein flexibility, which is a crutial shortcoming of current SBDD methods. From the data aspect, this paper utilizes Apobind to generate apo strutures from known holo structures. From the evaluation aspect, this paper presents a case study which analyzes the predicted structure in 5.4. For normal SBDD evaluation, this paper includes 2 datasets and adopts Glide scores.
2. For methodology, this paper respects characteristics of different modalities, and utilizes continuous, Riemanian, torus, and discrete flow matching for different modalities. Benefits from optimal transport path of flow matching, this paper achieves quite surprising sampling efficiency.
3. For technical innovation, this paper proposes, 1. data augmentation, 2. a good composition of geometric NN modules, both of which boost the performance.
4. The results look good. Congratulations, this paper makes it work.

**Weaknesses:**

This paper focuses on protein flexibility for SBDD. However, the evaluation (except for a case study in 5.4) mainly follows normal SBDD, while the readers may concern more about: 1. Quality of Apobind-generated apo structures and model-generated holo structures, which is not fully discussed and evaluated, 2. Why protein flexibility can help improve affinity for fixed holo structures? Thus, the reviewer concerns the thoroughness and presentation of the experiments. And the reviewer encourages the authors to elaborate on further evaluation and more discussion specific to protein-flexible SBDD.
I will raise my score if more flexible SBDD evaluations and discussions are presented.

**Questions:**

1. The reviewer believes FlexSBDD may generate different holo structures than ground-truth (GT). 1. The generated molecules should have higher affinities on generated holos than GT holos, right? 2. How different are the generated and GT holos? 3. If the GT ligand is provided, can the model generate accurate GT holos?
2. In Table 2, FlexSBDD shows much better bond length distribution than other baselines. How is bond generated? And why is it so good? And why is bond angles (Table 5) not much better?
3. In Table 3, why data augmentation boosts so much (and the most) on performance? It's hard to understand the connection between apo structures and affinity in fixed holos.
4. Affinity and other metrics have strong correlation with atom numbers. Please include atom numbers in Table 1 for fairer comparison.
I will raise my score if the questions are properly answered.

**Limitations:**

see weakness above

---

> ### Author Rebuttal · Authors · 2024-08-05
>
> We thank the reviewer for the appreciation and valuable comments! We hope our following responses can properly address your questions.
>
> **Comment 1**: Quality of Apobind-generated apo structures and model-generated holo structures, which is not fully discussed and evaluated
>
> **Response 1**: Thanks for the question! We use self-consistency TM (scTM) scores to evaluate the quality of apo/holo structures. The average scTM of apo structures from Apobind is 0.957 and the model generated holo is 0.964. We have a brief discussion in lines 276-282. We will make the discussion clearer in the revised paper.
>
> **Comment 2**: Why protein flexibility can help improve affinity for fixed holo structures? Thus, the reviewer concerns the thoroughness and presentation of the experiments. And the reviewer encourages the authors to elaborate on further evaluation and more discussion specific to protein-flexible SBDD.
>
> **Response 2**: Thanks for the valuable comment! According to the induced fit theory in biochemistry, proteins are flexible structures that undergo structural changes upon ligand binding, leading to enhanced interactions and binding affinity. Technically, modeling protein flexibility can help reduce steric clashes and adjust structure to establish more protein-ligand interactions such as hydrogen bonds to improve affinity. During rebuttal, we have done a series of further evaluations **including DFG-in/out structure prediction (uploaded pdf), distinct conformational search with respect to time/space scale (response 3), and comparing FlexSBDD with SBDD+flexible docking (response 4 to reviewer jwrm)**. These flexible SBDD specific evaluations shows the effectiveness of FlexSBDD. We will include these results and discussions into our final version.
>
> **Comment 3**: The reviewer believes FlexSBDD may generate different holo structures than ground-truth (GT). 1. The generated molecules should have higher affinities on generated holos than GT holos, right? 2. How different are the generated and GT holos? 3. If the GT ligand is provided, can the model generate accurate GT holos?
>
> **Response 3**: Thanks for the insightful comments!
> **Firstly**, it is true that the generated molecules have higher affinities on generated holos than GT holos. For example, the Avg. Vina Dock on generated holos is -9.12 while on GT holos is only -8.78. **Secondly**, the average RMSD between the generated and GT holos is 0.895, indicating the generated and GT holos are largely aligned.
> **Finally**, if the GT ligand is provided, FlexSBDD can generate accurate holos close to GT holos.
> We perform a comprehensive quantitative study on proteins with DFG-in/out confirmations to evaluate whether FlexSBDD can perform conformational searches on proteins with substantial structure variability. As shown in the uploaded pdf, the majority of the predicted protein structures show a lower relative pocket RMSD (better) compared to the initial ones, verifying FlexSBDD’s strong capability for ligand-specific conformational search.
>
> **Comment 4**: In Table 2, FlexSBDD shows much better bond length distribution than other baselines. How is bond generated? And why is it so good? And why is bond angles (Table 5) not much better?
>
> **Response 4**: In FlexSBDD, the bonds are generated with post-processing similar to TargetDiff. The powerful flow matching framework and well-designed model architecture contribute to the better bond length distribution. As for the bond angles, FlexSBDD still achieves competitive performance. The advantage over baselines methods may not be that much because bond angles are more complicated and FlexSBDD does not explicitly learn the bond angles (the bond representation are learned and updated in FlexSBDD). It will be our future work to design more powerful model architecture to learn the representations of bond angles.

---

> ### Author Response · Authors · 2024-08-05
> **Further Response to Reviewer 7REV**
>
> **Comment 5**: In Table 3, why data augmentation boosts so much (and the most) on performance? It's hard to understand the connection between apo structures and affinity in fixed holos.
>
> **Response 5**: Thanks for the insightful comment! Data augmentation plays an important role in FlexSBDD. As indicated in lines 186-195 of the submitted paper, we take apo-holo structure pairs for training. In each training iteration, we sample the apo structures $\mathcal{C}_0$ and holo-structures $\mathcal{C}_1$ and interpolate to obtain $\mathcal{C}_t$, i.e., FlexSBDD is supervised to learn the protein structural changes and the apo data play critical roles. However, existing apo-holo data pairs from Apobind are quite limiting (~10K data points) and directly training FlexSBDD on it leads to severe overfitting according to our experiments. The low-quality of the generated protein structure would directly lead to inferior Vina scores because of steric clashes and few protein-ligand interactions. **Therefore, we propose to use data augmentation to increase the training dataset size, cover more diverse apo-holo transition paths, and boost FlexSBDD’s generalization capability.** The ablation studies show that the data augmentation contributes a lot to FlexSBDD’s performance.
>
> We will include more detailed discussions and analysis of the role of data augmentation in our revised paper.
>
>
> **Comment 6**: Affinity and other metrics have strong correlation with atom numbers. Please include atom numbers in Table 1 for fairer comparison.
>
> **Response 6**: Thanks for the constructive suggestion! In FlexSBDD, the number of ligand atoms is sampled from the reference dataset distribution. We report the average number of ligand atoms below. Generally, the average num of atoms of FlexSBDD is comparable to the reference and other baseline methods. DecompDiff has the most number of atoms. We will include these statistics into the final version.
>
> | Methods    | Reference | LiGAN | AR | Pocket2Mol | TargetDiff | DecompDiff | FlexSBDD|
> |------------|--------------------|------|------------------------|------|---------------|------|------|
> | Avg. Num of Atoms  | 22.8     | 19.9| 17.7   | 24.2| 29.4     | 23.0 |

---

### Official Review · Reviewer_Y45d · 2024-07-06

**Soundness:** 2
**Presentation:** 3
**Contribution:** 2
**Rating:** 5
**Confidence:** 5

**Summary:**

In this research paper, the authors introduce FlexSBDD, a novel model that employs flow matching for the generation of flexible protein-based molecules. Initially, the model sample a noisy ligand based on an empirical distribution. Subsequently, it conducts flow matching, performing on both geometric characteristics and atomic types. Through comprehensive experimentation, FlexSBDD has demonstrated superior performance compared to previous methodologies, including TargetDiff, Pocket2Mol, and DecomDiff.

**Strengths:**

1. FlexSBDD explores a new avenue, flexible structure-based conditional molecular generation, which is quite novel.
2. FlexSBDD demonstrates superior performance through well-recognized benchmarking and ablation studies, adhering to rigorous research practices.
3. FlexSBDD discussions have many chemical insights, which is commendable.

**Weaknesses:**

1. As discussed in the Dynamic-Bind, many flexible aspects of protein residues involve changes in backbone atoms, such as transitions from DFG-in to DFG-out conformations. The current implementation of FlexSBDD does not demonstrate its capability for conformational search in more rigorous settings, such as when the apo pocket exhibits substantial dissimilarity from the holo pocket.
2. Although FlexSBDD represents a commendable effort, it fails to address a fundamental issue in flexible-pocket generation: evolving the appropriate atomic number through the generation process. FlexSBDD initializes ligands from an empirical distribution without considering pocket structures, resulting in a somewhat stochastic generation process: sampling a large number of ligands may artificially expand the pocket, leading to an increase in biased docking scores. The illustration provided with the 1a2g example supports this observation.
3. The model architecture of FlexSBDD, which includes flow matching on various geometries, has been previously implemented in other works such as PPFlow, diminishing the architectural novelty of FlexSBDD.
4. The illustrations in Figure 4, including the structures from 4yhj and 1fmc, display unusual bond topologies in molecules generated by FlexSBDD, such as a ring containing two double bonds. This issue may stem from an oversight in bond modeling within the architecture.

**Questions:**

1. In the evaluation using Vina dock metrics, it remains unclear which protein structures were utilized for benchmarking. Could you specify whether the benchmarks employed the initial holo structures or those updated by FlexSBDD?
2. Could you elaborate on the statement made in lines 216-217? “We note that it is fair to compare FlexSBDD with other baseline methods as the additional 217 apo structures contain no ligand molecules and cannot be used by baselines for training.” I do not fully understand the points here.
3. The code provided with the submission appears to lack both training and inference components, which undermines the credibility of the reported results. Could you address this omission?
4. Regarding the prediction of side-chain conformations, it appears that the analysis is limited to the mean squared error (MSE) of chi angles without considering the orientation within the residue frame prediction. Could you discuss the rationale behind this methodological choice?

**Limitations:**

The limitations and broader impacts are well discussed in Appendix E of the main paper.

---

> ### Author Rebuttal · Authors · 2024-08-05
>
> We thank the reviewer for the appreciation and valuable comments!
>
> **Comment 1**: As discussed in the Dynamic-Bind, many flexible aspects of protein residues involve changes in backbone atoms, such as transitions from DFG-in to DFG-out conformations. The current implementation of FlexSBDD does not demonstrate its capability for conformational search in more rigorous settings, such as when the apo pocket exhibits substantial dissimilarity from the holo pocket.
>
> **Response 1**: Thanks for the insightful comment! FlexSBDD has the capability to model both the backbone and the sidechain structural changes. During rebuttal, we perform a comprehensive quantitative study on proteins with DFG-in/out confirmations to evaluate whether FlexSBDD can perform conformational searches on proteins with substantial structure variability. **As shown in the uploaded pdf (https://openreview.net/forum?id=4AB54h21qG&noteId=Jo1MWV179v), the majority of the predicted protein structures show a lower pocket RMSD compared to the initial ones**, verifying FlexSBDD’s strong capability for conformational search.
>
> **Comment 2**: Although FlexSBDD represents a commendable effort, it fails to address a fundamental issue in flexible-pocket generation: evolving the appropriate atomic number through the generation process. FlexSBDD initializes ligands from an empirical distribution without considering pocket structures, resulting in a somewhat stochastic generation process: sampling a large number of ligands may artificially expand the pocket, leading to an increase in biased docking scores. The illustration provided with the 1a2g example supports this observation.
>
> **Response 2**: Thanks for the insightful comment! In FlexSBDD, the number of ligand atoms is sampled from the reference dataset distribution.
> According to related works [1-2], the flexible pocket can adaptively adjust structures to accommodate ligand molecules with different sizes. Therefore, sampling ligand molecules with different sizes help explore different binding modes, e.g., discover cryptic pockets. Pre-determining the ligand atom numbers may restrict the diversity and novelty of the generated molecules.
> We also report the average number of ligand atoms below. Generally, the average num of atoms of FlexSBDD is comparable to the reference and other baseline methods. DecompDiff has more Avg. Num of Atoms than FlexSBDD.
> As for the case study examples, we select the best generated ligand molecules for each target protein and FlexSBDD and DecompDiff have roughly the same number of atoms in most of cases.
>
> | Methods    | Reference | LiGAN | AR | Pocket2Mol | TargetDiff | DecompDiff | FlexSBDD|
> |------------|--------------------|------|------------------------|------|---------------|------|------|
> | Avg. Num of Atoms  | 22.8     | 19.9| 17.7   | 24.2| 29.4     | 23.0 |
>
> There are also some other promising techniques such training neural networks to predict the number of ligand atoms. We will include the discussions in our revised paper.
>
> [1] Lu W, Zhang J, Huang W, et al. DynamicBind: Predicting ligand-specific protein-ligand complex structure with a deep equivariant generative model[J]. Nature Communications, 2024, 15(1): 1071.
> [2] Qiao Z, Nie W, Vahdat A, et al. State-specific protein–ligand complex structure prediction with a multiscale deep generative model[J]. Nature Machine Intelligence, 2024, 6(2): 195-208.
>
>
> **Comment 3**: The model architecture of FlexSBDD, which includes flow matching on various geometries, has been previously implemented in other works such as PPFlow, diminishing the architectural novelty of FlexSBDD.
>
> **Response 3**: Thanks for the comment! Both FlexSBDD and PPFlow are based on Riemannian Flow Matching [3,4] proposed by previous works. Different from PPFlow that focus on peptide design, we propose a flow-matching-based generative model FlexSBDD, capable of modeling protein flexibility while generating de novo 3D ligand molecules. To work well on the challenging flexible SBDD scenario, FlexSBDD has unique designs on sidechain flow matching, scalar-vector dual representation architecture, and training with data augmentation.
> We will cite and discuss more related works in our revised paper.
>
> [3] Yaron Lipman, Ricky TQ Chen, Heli Ben-Hamu, Maximilian Nickel, and Matt Le. Flow matching for generative modeling. arXiv preprint arXiv:2210.02747, 2022.
> [4] Chen R T Q, Lipman Y. Riemannian flow matching on general geometries[J]. arXiv preprint arXiv:2302.03660, 2023.
>
> **Comment 4**: The illustrations in Figure 4, including the structures from 4yhj and 1fmc, display unusual bond topologies in molecules generated by FlexSBDD, such as a ring containing two double bonds. This issue may stem from an oversight in bond modeling within the architecture.
>
> **Response 4**: In FlexSBDD, the bonds are modeled with scalar/vector edge representations in the protein-ligand graph. The details of graph construction, feature initialization, message passing, and feature/structure update are included in Appendix D. We will provide more detailed and clear description in our revised paper. Actually, rings containing two double bonds are common in drug molecules, such as Metronidazole [5], Omeprazole [6], Pyrantel Pamoate [7], and Celecoxib [8]. The sub-structure analysis (Sec. 5.3) further validates that FlexSBDD generates valid bond distance/angle distributions.
> In the future, the bond modeling can be further improved with e.g., generating the bond types along with FlexSBDD sampling.
>
> [5] FINEGOLD S M. Metronidazole[J]. Annals of Internal Medicine, 1980, 93(4): 585-587.
> [6] Maton P N. Omeprazole[J]. New England Journal of Medicine, 1991, 324(14): 965-975.
> [7] Rim H J, Won C Y, Lee S I. Anthelmintic effect of oxantel pamoate and pyrantel pamoate[J]. The Korean Journal of Parasitology, 1975, 13(2): 97-01.
> [8] Puljak L, Marin A, Vrdoljak D, et al. Celecoxib for osteoarthritis[J]. Cochrane Database of Systematic Reviews, 2017 (5).

---

> ### Author Response · Authors · 2024-08-05
> **Further Response to Reviewer Y45d**
>
> **Comment 5**: In the evaluation using Vina dock metrics, it remains unclear which protein structures were utilized for benchmarking. Could you specify whether the benchmarks employed the initial holo structures or those updated by FlexSBDD?
>
> **Response 5**: For the Vina scores of FlexSBDD, we use the updated protein structure. As for the other baselines, we follow previous works to use the target structure from the test set for evaluation.
>
> **Comment 6**: Could you elaborate on the statement made in lines 216-217? “We note that it is fair to compare FlexSBDD with other baseline methods as the additional 217 apo structures contain no ligand molecules and cannot be used by baselines for training.” I do not fully understand the points here.
>
> **Response 6**: Thanks for the detailed question! In FlexSBDD, we associate holo structures from training dataset (CrossDocked and Binding MOAD) with apo conformations from Apobind [3] to create apo-holo pairs for training. We want to note that the additional Apobind dataset does not contain protein-ligand structures (i.e., only protein structures) and cannot be used by baseline methods for training. The Apobind dataset employed by FlexSBDD will not bring data leakage or additional advantage. Therefore, it is fair to compare FlexSBDD with baseline methods. We will make the statement clearer in our revised paper.
>
> **Comment 7**: The code provided with the submission appears to lack both training and inference components, which undermines the credibility of the reported results. Could you address this omission?
>
> **Response 7**: Thanks for the valuable comment! We have uploaded the training and inference codes. We will open-source all the codes upon paper acceptance.
>
> **Comment 8**: Regarding the prediction of side-chain conformations, it appears that the analysis is limited to the mean squared error (MSE) of chi angles without considering the orientation within the residue frame prediction. Could you discuss the rationale behind this methodological choice?
>
> **Response 8**: Thanks for the detailed comment! We use the mean squared error (MSE) of chi angles to evaluate the prediction of sidechain conformations following previous works [9-11]. Generally, lower MSE indicate more precise sidechain structure prediction. In table 6 of the paper, we can observe that FlexSBDD achieves better performance in generating valid sidechain structures.
>
> During rebuttal, we perform additional analysis of the side chain prediction. For example, we follow DynamicBind to conduct a comprehensive analysis six distinct conformational changes across the picosecond level to millisecond level (molecular dynamics), each exemplified by a case from PDBbind. In the following table, we report Δpocket RMSD (including side chain and backbone) of DynamicBind and FlexSBDD, which measures the relative decrease in pocket RMSD (crystal structure as reference) compared with the AlphaFold structures. **A negative Δpocket RMSD indicates that the predicted aligns more closely with the crystal structure compared with the AlphaFold prediction.** We observe that FlexSBDD achieves competitive performance that improves the AlphaFold prediction to have lower pocket RMSD, even though it is not specifically designed for dynamic docking.
>
> | Methods    | 6QGF | 6PGO | 6N8X | 6UWV | 6ROT | 6S9X |
> |------------|--------------------|------|------------------------|------|---------------|------|
> | DynamicBind  | -0.669     | -1.140| -2.297    | -0.465| -2.327      | -5.245 |
> | FlexSBDD| -0.680    | -0.976| -1.159  | -0.504| -1.148   | -3.083 |
>
> [9] Zhang Y, Zhang Z, Zhong B, et al. Diffpack: A torsional diffusion model for autoregressive protein side-chain packing. Advances in Neural Information Processing Systems, 2023.
> [10] McPartlon M, Xu J. An end-to-end deep learning method for protein side-chain packing and inverse folding[J]. Proceedings of the National Academy of Sciences, 2023, 120(23): e2216438120.
> [11] Dong T, Yang Z, Zhou J, et al. Equivariant flexible modeling of the protein–ligand binding pose with geometric deep learning[J]. Journal of Chemical Theory and Computation, 2023, 19(22): 8446-8459.

---

### Official Review · Reviewer_jwrm · 2024-07-08

**Soundness:** 3
**Presentation:** 3
**Contribution:** 3
**Rating:** 7
**Confidence:** 4

**Summary:**

The author identified an important missing factor in current SBDD modeling, i.e. protein structural change upon binding, and proposed an E(3)-equivariant flow matching framework named FlexSBDD that jointly models protein flexibility and molecule generation. This paper augmented apo structures for each holo protein in the training set via structure relaxation, Rosetta repacking and random perturbations. FlexSBDD achieves SOTA binding affinities, QED and diversity benchmarked on CrossDocked2020 and Binding MOAD dataset, together with fewer clashes and more HB donors and acceptors.

**Strengths:**

- This paper is well-motivated and generally easy to follow (except for model architecture and training).
- The authors raise an important question for SBDD, i.e. holo-structures are an induced fit for ligand molecules, which is a novel contribution.
- The results on two benchmarks are convincing and highlight the importance of modeling protein flexibility.

**Weaknesses:**

- Typo: (Line 90-91) However, these methods can hardly [be] extended to the challenging de novo ligand generation, leaving it an unsolved problem.
- Typo: (Line 591) Protein Sturcture Analysis => Protein Structure Analysis
- Ablation studies suggest that the biggest performance gain come from data augmentation. However, I feel that augmentation shouldn't matter that much, since the whole pipeline only implicitly utilizes the protein structural change and the explicit outcome is the generated ligand itself. Could the authors explain the role of data augmentation in your method?
- Since apo (unbound) and holo (bound) state proteins are of great focus in this paper, it seems to me that more reasonable metrics for protein analysis would be based on some molecular dynamics, instead of scTM or something that are not guaranteed to output holo proteins for given ligand molecules.

**Questions:**

- How would FlexSBDD behave if applied to an inpainting scenario (with ground truth holo-protein)?
- For a fair comparison, I would recommend the authors to try some flexible docking tools that also take the protein flexibility into account, and see how the performances of FlexSBDD and other baselines change.
- The authors could elaborate a bit more on the evaluation of protein structures. For example, why is FlexSBDD superior to SOTA protein-ligand complex structure prediction method? Under what setting are they being evaluated and compared?

---

> ### Author Rebuttal · Authors · 2024-08-05
>
> We thank the reviewer for valuable comments and appreciation!
>
> **Comment 1**: Ablation studies suggest that the biggest performance gain come from data augmentation. However, I feel that augmentation shouldn't matter that much, since the whole pipeline only implicitly utilizes the protein structural change and the explicit outcome is the generated ligand itself. Could the authors explain the role of data augmentation in your method?
>
> **Response 1**: Thanks for the insightful comment! Data augmentation plays important roles in FlexSBDD and the protein structural changes are considered explicitly in the whole pipeline. As indicated in lines 186-195 of the submitted paper, we take apo-holo structure pairs for training. In each training iteration, we sample the apo structures $\mathcal{C}_0$ and holo-structures $\mathcal{C}_1$ and interpolate to obtain $\mathcal{C}_t$, i.e., FlexSBDD is supervised to learn the protein structural changes and the data play critical roles. However, existing apo-holo data pairs from Apobind are quite limiting (~10K data points) and directly training FlexSBDD on it leads to severe overfitting according to our experiments. The low-quality of the generated protein structure would directly lead to inferior Vina scores because of steric clashes and few protein-ligand interactions. **Therefore, we propose to use data augmentation to increase the training dataset size, cover more apo-holo transition paths, and boost FlexSBDD’s generalization capability.** The ablation studies show that the data augmentation contributes a lot to FlexSBDD’s performance.
>
> We will include more detailed discussions and analysis of the role of data augmentation in our revised paper.
>
> **Comment 2**: Since apo (unbound) and holo (bound) state proteins are of great focus in this paper, it seems to me that more reasonable metrics for protein analysis would be based on some molecular dynamics, instead of scTM or something that are not guaranteed to output holo proteins for given ligand molecules.
>
> **Response 2**: Thanks for the constructive suggestion! We follow DynamicBind to conduct a comprehensive analysis six distinct conformational changes **across the picosecond level to millisecond level (molecular dynamics)**, each exemplified by a case from PDBbind. In the following table, we report Δpocket RMSD of DynamicBind and FlexSBDD, which measures the relative decrease in pocket RMSD (crystal structure as reference) compared with the AlphaFold structures. **A negative Δpocket RMSD indicates that the predicted aligns more closely with the crystal structure compared with the AlphaFold prediction.** We observe that FlexSBDD achieves competitive performance although it is not specifically designed for dynamic docking.
>
> | Methods    | 6QGF | 6PGO | 6N8X | 6UWV | 6ROT | 6S9X |
> |------------|--------------------|------|------------------------|------|---------------|------|
> | DynamicBind  | -0.669     | -1.140| -2.297    | -0.465| -2.327      | -5.245 |
> | FlexSBDD| -0.680    | -0.976| -1.159  | -0.504| -1.148   | -3.083 |
>
>
> **Comment 3**: How would FlexSBDD behave if applied to an inpainting scenario (with ground truth holo-protein)?
>
> **Response 3**: Thanks for the question! We agree it is a good comparison to apply FlexSBDD to the inpainting scenario (with ground truth holo-protein), which is the same setting with the baseline methods (LiGAN, AR, Pocket2Mol, TargetDiff, and DecompDiff). We show the results below and observe that FlexSBDD still behaves well in the inpaint setting, showing the generalizability and flexibility of FlexSBDD’s architecture.
>
> | Methods    | Vina Score (↓) Avg. | Med. | Vina Min (↓) Avg. | Med. | Vina Dock (↓) Avg. | Med. | High Affinity (↑) Avg. | Med. | QED (↑) Avg. | Med. | SA (↑) Avg. | Med. | Diversity (↑) Avg. | Med. |
> |------------|---------------------|------|--------------------|------|--------------------|------|------------------------|------|---------------|------|-------------|------|--------------------|------|
> | **Reference**   | -6.36               | -6.46| -6.71              | -6.49| -7.45              | -7.26| -                      | -    | 0.48          | 0.47 | 0.73        | 0.74 | -                  | -    |
> | LiGAN       | -                   | -    | -                  | -    | -6.33              | -6.20| 21.1%                  | 11.1%| 0.39          | 0.39 | 0.59        | 0.57 | 0.66               | 0.67 |
> | AR          | *-5.75*             | -5.64| -6.18              | -5.88| -6.75              | -6.62| 37.9%                  | 31.0%| 0.51          | 0.50 | {0.63}      |{0.63}| 0.70               | 0.70 |
> | Pocket2Mol  | -5.14               | -4.70| -6.42           | -5.82| -7.15              | -6.79| 48.4%                  | 51.0%| **0.56**      | **0.57** | **0.74**  | **0.75** | 0.69            | *0.71*  |
> | TargetDiff  | -5.47               | *-6.30*| -6.64            | -6.86| -7.80              | -7.91| 58.1%                  | 59.1%| 0.48          | 0.48 | 0.58        | 0.58 | 0.72               | *0.71*  |
> | DecompDiff  | -5.67               | -6.04 | *-7.04*           | *-6.91*| *-8.39*          | *-8.43*| *64.4%*             | *71.0%*| 0.45         | 0.43 | 0.61        | 0.60 | 0.68               | 0.68  |
> | FlexSBDD (inpaint)| **-6.69**           | **-7.16**| **-8.24**      | **-8.50**| **-9.06**      | **-9.12**| **75.9%**         | **82.1%**| **0.58**    | **0.58** | *0.70*    | *0.71* | **0.74**          | **0.72**  |
>
> - **Bold**: Best results
> - *Italic*: Second best

---

> ### Author Response · Authors · 2024-08-05
> **Further Response to Reviewer jwrm**
>
> **Comment 4**: For a fair comparison, I would recommend the authors to try some flexible docking tools that also take the protein flexibility into account, and see how the performances of FlexSBDD and other baselines change.
>
> **Response 4**: Thanks for the valuable suggestion! In the following table, **we combine the top-2 SBDD baselines with DynamicBind [51], the flexible docking to compare with FlexSBDD.** Specifically, after generating the ligand molecules with TargetDiff/DecompDiff, we further dock the ligand to the target protein with DynamicBind. For fair comparison, we show the results of Vina Dock, QED, SA, and Diversity below. We observe that applying flexible docking as post-processing indeed improve the Vina Dock of baselines. FlexSBDD still achieves the best results as an end-to-end generative model for de novo ligand generation while adjusting protein structures. We will include these new results and discussions in our revised paper.
>
> | Methods    | Vina Dock (↓) Avg. | Med. | High Affinity (↑) Avg. | Med. | QED (↑) Avg. | Med. | SA (↑) Avg. | Med. | Diversity (↑) Avg. | Med. |
> |------------|--------------------|------|------------------------|------|---------------|------|-------------|------|--------------------|------|
> | TargetDiff  | -8.17              | -8.25| 62.3%                  | 63.0%| 0.48          | 0.48 | 0.58        | 0.58 | 0.72               | *0.71*  |
> | DecompDiff  | *-8.89*            | *-8.97*| *69.1%*             | *74.5%*| 0.45         | 0.43 | 0.61        | 0.60 | 0.68               | 0.68  |
> | **FlexSBDD**| **-9.12**          | **-9.25**| **78.5%**         | **84.2%**| **0.58**    | **0.59** | *0.69*    | *0.73* | **0.76**          | **0.75**  |
>
> - **Bold**: Best results
> - *Italic*: Second best
>
> [51] Wei Lu, Ji-Xian Zhang, Weifeng Huang, Ziqiao Zhang, Xiangyu Jia, Zhenyu Wang, Leilei Shi, Chengtao Li, Peter Wolynes, and Shuangjia Zheng. Dynamicbind: Predicting ligand-specific protein-ligand complex structure with a deep equivariant generative model. 2023.
>
> **Comment 5**: The authors could elaborate a bit more on the evaluation of protein structures. For example, why is FlexSBDD superior to SOTA protein-ligand complex structure prediction method? Under what setting are they being evaluated and compared?
>
> **Response 5**: Thanks for the valuable suggestion! In Appendix B.3, we evaluate the sidechain prediction of FlexSBDD and compare it with SOTA protein-ligand complex structure prediction method NeuralPlexer. **We use the inpaint setting for experiments: the NeuralPLexer model is asked to jointly predict the structure for a cropped spherical region within 6.0 Å of any ligand atom by inpainting all the amino acid and ligand atomic coordinates from scratch; FlexSBDD generate the ligand structure and updates the apo structure to holo.** To evaluate the validity of sidechain structure, we compute the Mean Absolute Error (MAE) of sidechain angles (degrees) of the inpainting region. In Table 6, we observed that FlexSBDD achieves better results on MAE. This could be attributed to the advance flow matching framework, scalar-vector dual representation architecture, and data augmentation strategy. We will include more details of the experimental settings and discussions in the revised version.
>
> **Comment 6**: Typos: (Line 90-91) However, these methods can hardly [be] extended to the challenging de novo ligand generation, leaving it an unsolved problem. (Line 591) Protein Sturcture Analysis => Protein Structure Analysis
>
> **Response 6**: Thanks for the detailed comments! We have corrected our typos and will submit the updated paper in the final version.

---

> > ### Comment · Reviewer_jwrm · 2024-08-11
> >
> > Thanks for the authors' detailed response. It addressed all my concerns. I have raised my score to 7 in hopes that this paper gets accepted.

---

> > > ### Author Response · Authors · 2024-08-11
> > > **Thanks for your support!**
> > >
> > > Dear Reviewer,
> > >
> > > Thanks for your support! We are glad that our response addressed all your concerns.
> > >
> > > Bests,
> > > Authors

---

### Official Review · Reviewer_QC25 · 2024-07-12

**Soundness:** 3
**Presentation:** 3
**Contribution:** 3
**Rating:** 5
**Confidence:** 3

**Summary:**

In this paper, the authors focus on the flexible protein setting in the structure-based drug design task. They propose a method named FlexSBDD, which is based on flow matching and utilizes E(3)-equivariant neural networks. The experiments show the advantages of the proposed FlexSBDD.

**Strengths:**

1. This paper focuses on an interesting setting where the proteins are flexible.
2. The presentation is good, and the paper is well-organized.

**Weaknesses:**

1. The method does not explicitly model the interaction between the ligand and the protein, especially the pocket. The authors might consider building an external interaction graph between the residues in the pocket and the atoms of the ligand.
2. I would like to see more focused results on the binding interface.

**Questions:**

See the weaknesses. Additionally, I noticed a work [1] that is closely related to this paper, also employing flow matching. However, the authors have not cited or discussed the differences between their work and this one.


**Minor Concern:**

Line 240 "Table.": Since "Table" is written in full without abbreviation, there is no need to add a period after "Table". This issue occurs in multiple places.



**Reference:**

[1] Schneuing, Arne, et al. "Towards Structure-based Drug Design with Protein Flexibility." ICLR 2024 Workshop on Generative and Experimental Perspectives for Biomolecular Design.

**Limitations:**

The authors have discussed the limitations of FlexSBDD.

---

> ### Author Rebuttal · Authors · 2024-08-05
>
> We thank the reviewer for the valuable comments!
>
> **Comment 1**: The method does not explicitly model the interaction between the ligand and the protein, especially the pocket. The authors might consider building an external interaction graph between the residues in the pocket and the atoms of the ligand.
>
> **Response1**: As indicated in lines 666-668 of the submitted paper, in FlexSBDD, we represent the protein pocket-ligand complex as a k-nearest neighbor (KNN) graph in which nodes represent protein residues or ligand atoms and each node is connected to its k-nearest neighbors. Therefore, the interactions between the ligand and the protein are considered as edge representations in the constructed protein-ligand graph.
>
> We can also incorporate an additional interaction graph to emphasize protein-ligand interactions. For instance, we may train interaction/binding affinity predictors based on protein-ligand graphs and use these trained predictors to guide the sampling process of FlexSBDD, similar to the approach described in [1]. These only require minor modifications to the FlexSBDD framework. We will include these additional discussions in our revised paper.
>
> [1] Qian H, Huang W, Tu S, et al. KGDiff: towards explainable target-aware molecule generation with knowledge guidance[J]. Briefings in Bioinformatics, 2024, 25(1): bbad435.
>
> **Comment 2**: I would like to see more focused results on the binding interface.
>
> **Response 2**: Thanks for the valuable comments! Besides the common benchmark metrics adopted by previous SBDD papers such as Vina scores, QED, and SA, we also focus on investigating the interactions on the binding interface. **In Figure 3, we consider steric clashes, hydrogen bonds, and hydrophobic interactions at the protein ligand binding interface.** Steric clashes happens when two neutral atoms come into closer proximity than the combined extent of their van der Waals radii, indicating energetically unfavorable and physically unrealistic structures. Hydrogen bonds (HBs) and Hydrophobic interactions are polar interactions that significantly contribute to the binding affinity between proteins and ligands.
> We observe that FlexSBDD can generate ligands introducing fewer clashes and more favorable interactions. For example, the average steric clashes for DecompDiff (baseline) and FlexSBDD are 6.43 and 1.39 respectively. The average number of HB Acceptors for DecompDiff (baseline) and FlexSBDD are 1.18 and 1.96 respectively. These results indicate that FlexSBDD can adaptively adjust protein and ligand conformations to reduce clashes and increase favorable protein-ligand interactions.
>
> These analysis are included in lines 257-268 of the paper. We will include more comprehensive analysis in the revised paper.
>
> **Comment 3**: I noticed a work [2] that is closely related to this paper, also employing flow matching. However, the authors have not cited or discussed the differences between their work and this one.
> [2] Schneuing, Arne, et al. "Towards Structure-based Drug Design with Protein Flexibility." ICLR 2024 Workshop on Generative and Experimental Perspectives for Biomolecular Design.
>
> **Response 3**: Thanks for mentioning the related paper. The ICLR24 workshop paper represents a pioneering work to consider structure-based drug design with protein flexibility. However, it only considers side chain flexibility while keeping backbone atoms fixed. The authors said the full-fledged induced-fit requires backbone movement modeling. Moreover, it only trains on protein-ligand binding complex structures and did not consider apo-holo structure pairs to learn structural transitions. Therefore, the performance of FlexFlow from the ICLR24 workshop is quite limited, even worse than its counterpart without flexible sidechain modeling.
>
> In comparison, FlexSBDD is able to model both sidechain and backbone flexibility of the protein while generating de novo ligand molecules. To better learn the transition between apo and holo structures, we associate data from the training set with apo structures from Apobind. We also create synthetic apo conformations with OpenMM relaxation and Rosetta repacking. As for the performance, FlexSBDD not only achieves state-of-the-art performance on benchmark datasets (e.g., -9.12 Avg. Vina Dock score), but also learns to adjust the protein structure to increase favorable interactions (e.g., 1.96 Avg. Hydrogen bond acceptors) and decrease steric clashes.
>
> We will cite and include the discussions in our revised paper.
>
> **Comment 4**: Line 240 "Table.": Since "Table" is written in full without abbreviation, there is no need to add a period after "Table". This issue occurs in multiple places.
>
> **Response 4**: Thanks for the valuable comments! We have corrected the typos in our paper and would like to update the submitted paper in the final version.

---

> > ### Comment · Reviewer_QC25 · 2024-08-13
> >
> > Thanks for your response. I will keep my original score.

---

> > > ### Author Response · Authors · 2024-08-13
> > > **Thanks for your response!**
> > >
> > > Dear Reviewer,
> > >
> > > Thanks for your valuable suggestions and support! We will include the above discussions in our revised paper. Thanks!
> > >
> > > Bests,
> > > Authors

---

### Author Rebuttal · Authors · 2024-08-05

Thanks for the insightful comments and appreciation from all the reviewers!

FlexSBDD has the capability to model both the backbone and the sidechain structural changes. During rebuttal, we perform a comprehensive quantitative study on proteins with DFG-in/out confirmations to evaluate whether FlexSBDD can perform conformational searches on proteins with substantial structure variability. As shown in the uploaded pdf, the majority of the predicted protein structures show a lower relative pocket RMSD (better) compared to the initial ones, verifying FlexSBDD’s strong capability for ligand-specific conformational search.

As for the specific questions, our responses are included in the following paragraphs. Thanks!

---

### Public Comment · ~Kiwoong_Yoo1 · 2025-04-03
**Code implementation release date?**

When would the code implementation be ready? The code repository seems empty. Thanks

---

### Decision · Program_Chairs · 2024-09-25

**Decision:**

Accept (poster)

**Comment:**

This work explores the use of dynamic information in molecular design, introducing a method called FlexSBDD based on flow matching and utilizing E(3)-equivariant neural networks. Experimental results demonstrate that the model achieves state-of-the-art performance across multiple benchmarks. After a thorough rebuttal, the reviewers found this work to be universally innovative and timely, given the growing capabilities in exploring protein dynamics for drug discovery. Despite some minor concerns, this is a good paper that I recommend for acceptance to NeurIPS.